# CURVE: Learning Causality-Inspired Invariant Representations for Robust Scene Understanding via Uncertainty-Guided Regularization

Yue Liang [1]   Jiatong Du [2]   Ziyi Yang [2]   Yanjun Huang [2]   Hong Chen [3]

## Abstract

Scene graphs provide structured abstractions for scene understanding, yet they often overfit to spurious correlations, severely hindering out-of-distribution generalization. To address this limitation, we propose CURVE, a causality-inspired framework that integrates variational uncertainty modeling with uncertainty-guided structural regularization to suppress high-variance, environment-specific relations. Specifically, we apply prototype-conditioned debiasing to disentangle invariant interaction dynamics from environment-dependent variations, promoting a sparse and domain-stable topology. Empirically, we evaluate CURVE in zero-shot transfer and low-data sim-to-real adaptation, verifying its ability to learn domain-stable sparse topologies and provide reliable uncertainty estimates to support risk prediction under distribution shifts. The code is released at https://github.com/ly3580/CURVE.

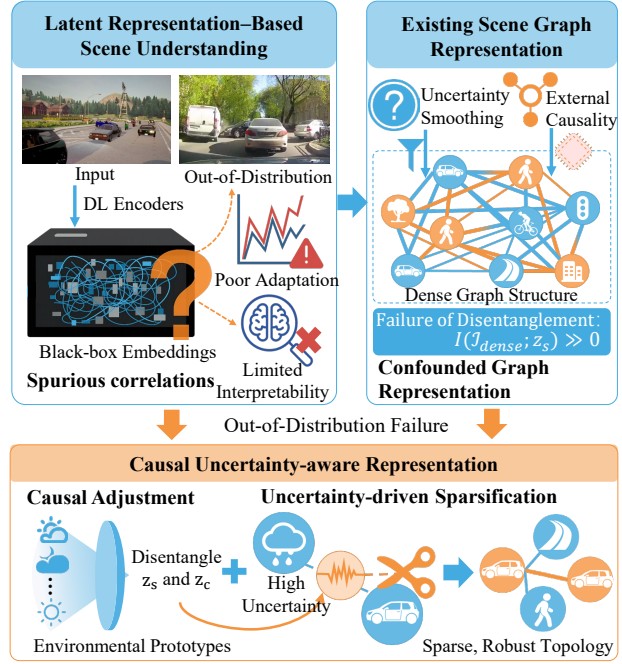

*Figure 1.* **CURVE mitigates OOD failures by counteracting environment-driven dense edges.** The top panels illustrate OOD failures in latent/dense graph approaches, as dense connectivity induces environment-conditioned edges, encouraging shortcut reliance on background co-occurrences. The bottom panel demonstrates our method using soft causal backdoor adjustment and uncertainty-driven sparsification to separate factors and produce a sparse, robust graph.

## 1. Introduction

Achieving robust generalization in safety-critical environments remains a fundamental challenge. While deep learning approaches achieve remarkable in-distribution success by mapping features into high-dimensional, implicit latent spaces, they notoriously suffer from brittleness in out-of-distribution (OOD) scenarios(Li et al., 2024a). This is primarily because models often rely on spurious correlations, such as associating risk with specific weather textures rather than actual agent dynamics(Li et al., 2025). While these statistical shortcuts might hold in the training set, they break

down in novel environments where such associations no longer hold.

Graph-structured representations offer a powerful solution to statistical shortcuts. Unlike unstructured latent embeddings, scene graphs explicitly decouple agents from the environment, theoretically preventing overfitting to background noise(Zhang et al., 2024b; Hu et al., 2025; Mo et al., 2025). However, current approaches often fail to realize this promise. By treating all connections as important, they generate dense topologies that confuse physical rules with environmental details, as shown in Figure 1. Consequently, they regress to the same limitations of unstructured latent representations, reducing the explicit graph to a noisy, en-

[1]Shanghai Research Institute for Intelligent Autonomous Systems, Tongji University, Shanghai, China [2]College of Automotive and Energy Engineering, Tongji University, Shanghai, China [3]College of Electronics and Information Engineering, Tongji University, Shanghai, China. Correspondence to: Yanjun Huang <yanjun_huang@tongji.edu.cn>, Hong Chen <chenhong2019@tongji.edu.cn>.

*Proceedings of the 43rd International Conference on Machine Learning*, Seoul, South Korea. PMLR 306, 2026. Copyright 2026 by the author(s).

tangled black box(Li et al., 2022; Yao et al., 2025).

This degradation is driven by two key factors. First, prior work uses uncertainty to smooth predictions under a fixed (often dense) topology, instead of pruning unreliable edges. Existing methods like PC-SGG(Nag et al., 2025) and FloCoDe(Khandelwal, 2023) attempt to mitigate sensor noise by generating prediction sets or enforcing temporal consistency to smooth out fluctuations. Similarly, HiKER-SGG(Zhang et al., 2024a)utilizes hierarchical knowledge graphs as priors to enhance robustness against impaired observations. While effective for stability, these approaches operate on fixed topologies, treating uncertainty merely as a value fluctuation to be smoothed. Crucially, high uncertainty is empirically associated with environment-sensitive or potentially spurious relations. By smoothing these fluctuations rather than pruning the connections, current methods inadvertently preserve the very statistical shortcuts that cause model collapse in OOD environments.

Second, existing approaches rely heavily on statistical co-occurrences, treating causality merely as an external patch rather than an intrinsic learning objective. While recent works like SRD-SGG (Nguyen et al., 2025), RcSGG (Sun et al., 2025) and MCCM (Liu et al., 2025) attempt to incorporate causal reasoning, they typically rely on handcrafted priors or post-hoc adjustments to correct specific biases. This superficial integration fails to fundamentally disentangle the underlying physical mechanisms from environmental context. Consequently, the representations remain entangled, leaving the model prone to confounding stable dynamics with domain-specific biases.

To this end, we propose **CURVE**, a Causal Uncertainty-aware Representation for Vehicle Environments. Unlike deterministic approaches, CURVE establishes a variational framework that leverages uncertainty not just for prediction, but as a proxy for spuriousness. Crucially, we overcome the limitations of continuous confounding by developing a prototype-driven soft causal intervention module. This module actively discretizes the latent space to facilitate tractable backdoor adjustment, forcing the model to rectify environmental biases. This mechanism explicitly encourages the separation of domain-stable interaction factors from environment-dependent correlations, inducing a sparse, robust topology grounded in physical reality.

Our contributions are as follows:

(1) We propose a probabilistic framework that integrates variational uncertainty with causality-inspired regularization. It repurposes uncertainty from a metric for denoising to a criterion for graph sparsification and robust topology induction.

(2) We construct a prototype-driven adjustment mechanism to learn causal-invariant representations, performing a soft,

feature-space backdoor-inspired adjustment to separate invariant interaction factors from confounding influences.

(3) We rigorously validate CURVE across low-data regimes and zero-shot transfer settings. Experiments confirm its superior OOD generalization, effective structural sparsification, and safety-critical reliability compared to state-of-the-art baselines.

## 2. Related Work

### 2.1. Scene Graphs for Representation and Learning

SGs provide a structured and interpretable representation of a driving scene by modeling objects as nodes and their interactions as edges. They offer a compact abstraction of spatial, semantic, and relational properties, enabling high-level reasoning tasks like risk assessment and collision prediction. Early SGs are primarily rule-based, built from hand-crafted semantic rules, geometric relationships, and map topology(Malawade et al., 2022). While such representations are highly interpretable, they lack flexibility, restricting them to specific tasks and data domains.

Recent approaches learn SGs directly from data by leveraging deep detectors, relational encoders, and graph neural networks to infer nodes and their interactions. These models facilitate large-scale relational reasoning and power tasks such as collision prediction and risk estimation. For instance, RS2G(Wang et al., 2024a) learns road scene graphs from data using a Transformer–VAE architecture, HDGT(Jia et al., 2023) models driving scenes as heterogeneous graphs for multi-agent prediction, and ISG(Zhang et al., 2025) jointly encodes dynamic agents and static context via separate Transformers. However, purely data-driven SGs rely on statistical correlations rather than causal relationships, making them vulnerable to spurious correlations, sensor perturbations, and domain shifts.

More recent efforts explore knowledge-guided SGs by incorporating semantic priors, structural constraints, interaction norms, or causal reasoning principles. For example, HKTSG(Zhou et al., 2024) integrates a human-like hierarchical cognition process into the data-driven learning paradigm, effectively leveraging both domain-general and domain-specific knowledge. Lu et al. (2023) further exploit prior knowledge and causal inference to guide dynamic SG generation. These methods aim to enhance interpretability and generalization, moving beyond purely data-driven, correlation-based models. However, purely data-driven SGs typically generate dense topologies learned directly from noisy observations. Lacking constraints to distinguish causation from correlation, they tend to overfit to these spurious patterns.

## 2.2. Causal and Knowledge-guided Scene Understanding

While Scene Graphs provide a structured representation of the environment, causal and knowledge-guided methods capture the underlying logic and interaction dynamics. Early approaches mainly rely on symbolic rules and logic to encode domain-specific knowledge and relational structure(Bagschik et al., 2018). Building on this, works like COSI(Halilaj et al., 2021) organize scene entities and relations into structured graphs for situation comprehension.

To overcome the rigidity of purely symbolic models, recent methods integrate deep perceptual encoders with causal reasoning or knowledge-guided constraints. For example, DSceneKG(Wickramarachchi et al., 2024) generates knowledge graphs of driving scenes, forming a foundation for neurosymbolic AI. MCAM(Cheng et al., 2025) designs a causal analysis module based on a directed acyclic graph (DAG) to dynamically model driving scenarios, while DDLoss(Peng et al., 2024) introduces a causality-guided loss function to more effectively handle imbalanced scene classification. However, these methods predominantly rely on fixed causal priors or deterministic graph structures, overlooking the stochastic nature of perceptual observations. Consequently, they lack a mechanism to leverage uncertainty as a filter, failing to explicitly disentangle invariant causal dynamics from transient environmental noise. Therefore, we introduce a prototype-driven, feature-space backdoor-inspired adjustment to separate invariant interaction factors from confounding environmental influences.

## 3. Methodology

### 3.1. Problem Formulation

**Structural Causal Model (SCM).** We describe the scene graph generation process of autonomous driving through a causality-inspired generative model, following the intuition of structural causal mechanisms $\mathcal{M} := \langle \mathcal{E}, \mathcal{Z}, \mathcal{G} \rangle$. Let $e \in \mathcal{E}$ denote the environment domain, which acts as an exogenous variable. The state of the world is characterized by two distinct sets of latent variables $\mathcal{Z} = \{z_c, z_s\}$ (Morioka & Hyvarinen, 2024).

Invariant Causal Factors $z_c$ represent the physical reality of the scene, such as object geometry, kinematics, and spatial topology. We assume the mechanism generating $z_c$ is invariant across domains, satisfying $z_c \perp\!\!\!\perp e$. Spurious confounding influences $z_s$ represent domain-specific attributes, such as illumination, weather textures, and sensor noise. Their distribution is fundamentally dependent on the environment, denoted as $z_s \not\!\perp\!\!\!\perp e$.

The raw observation $X$ is generated by a non-linear mixing function $\mathcal{G}$ that maps the latent factors to the pixel space:

$$X := \mathcal{G}(z_c, z_s). \tag{1}$$

In standard data-driven approaches, the learned representation $\hat{Z}$ typically captures both factors, failing to distinguish between the stable signal $z_c$ and the transient noise $z_s$.

From a causal perspective, the learned representation $\hat{Z}$ is obtained from $X$ via a parametric encoder and therefore inherits information from both $z_c$ and $z_s$. Under this modeling assumption, the downstream prediction target $Y$ is primarily governed by invariant physical factors $z_c$. However, since $z_s$ influences $\hat{Z}$ through the observation $X$, it may induce spurious statistical associations between $\hat{Z}$ and $Y$ that do not reflect the true causal mechanism. Under this causal graph, $z_s$ acts as a confounder between the learned representation and the prediction outcome.

**The Ideal Objective.** Our goal is to learn a scene graph encoder $\Phi$ that maps an observation $x$ to a structured interaction representation $\mathcal{I}$. Ideally, $\mathcal{I} = \Phi(x)$ should maximize information retention regarding the causal factors $z_c$ while strictly compressing out the spurious environment-dependent factors, minimizing the information leakage from $z_s$:

$$\max_{\Phi} I(\mathcal{I}; z_c) \quad \text{s.t.} \quad I(\mathcal{I}; z_s) \approx 0. \tag{2}$$

where $I(\cdot; \cdot)$ denotes the mutual information. Under the above causal structure, achieving invariance to $z_s$ can be viewed as identifying the causal effect of the learned representation on the prediction outcome while blocking spurious backdoor paths induced by environment-dependent factors.

**Practical Surrogate Objective.** Ideally, the representation $\mathcal{I}$ should perfectly isolate the invariant causal factors $z_c$ from the spurious correlations $z_s$, but impossible in practice due to the theoretical non-identifiability of latent factors. Instead, we leverage environmental invariance to guide the representation learning.

We assume that the causality-consistent factors $z_c$ are governed by domain-invariant mechanisms, remaining stable despite distributional shifts. Consequently, this ideal objective can be translated into a robust transfer problem. We posit that the more stable the encoder $\Phi$ is across varying domains, the more closely the representation $\mathcal{I}$ aligns with the invariant causal factors $z_c$, thereby effectively disentangling it from the unstable spurious correlations $z_s$ (Varıcı et al., 2024; Morioka & Hyvarinen, 2024).

### 3.2. Variational Scene Graph Generation

As a prerequisite for disentangling $z_c$ and $z_s$, we first construct an entangled probabilistic representation $\mathcal{I}$ grounded

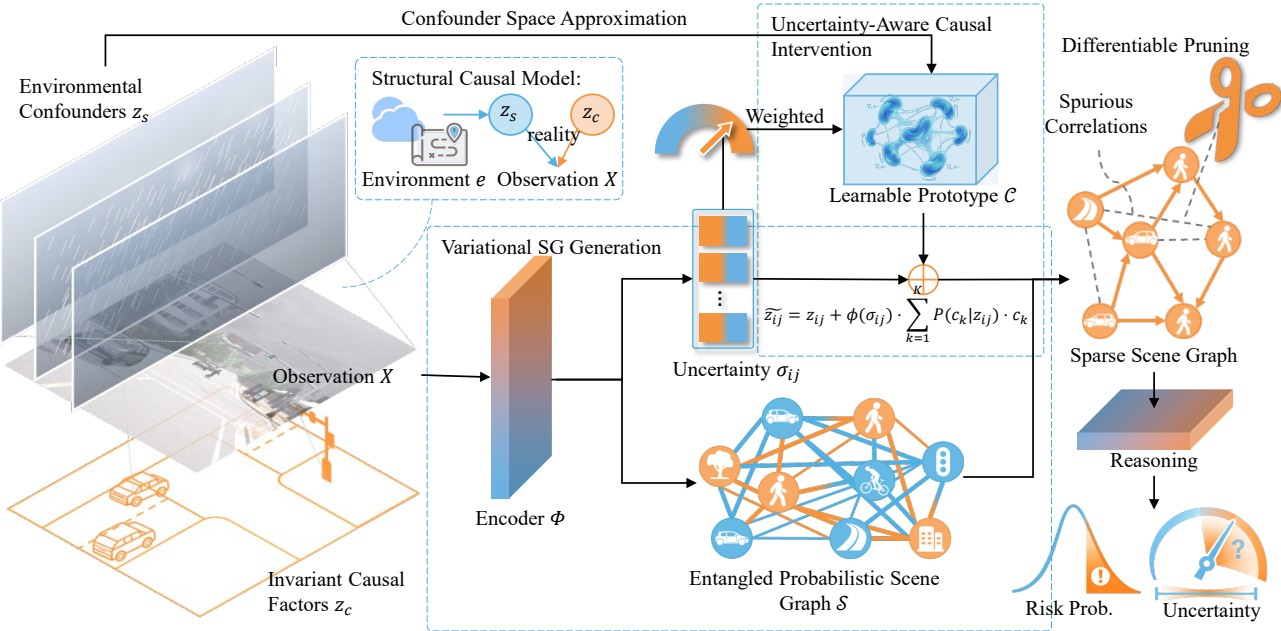

*Figure 2.* Overview of CURVE framework. Our approach disentangles invariant causal factors $z_c$ from environmental bias factors $z_s$. The core mechanism employs data-dependent uncertainty $\sigma_{ij}$ to reweight a causal intervention, leveraging a learnable prototype dictionary $\mathcal{C}$ to estimate and compensate for environment-conditioned bias. Finally, differentiable pruning eliminates spurious correlations to yield a sparse causality-consistent graph for robust reasoning and calibrated prediction.

on the explicit scene graph $\mathcal{S} = (\mathcal{V}, \mathcal{R})$. Instead of deterministic point estimates, we model $\mathcal{I}$ as a hierarchical distribution characterizing both detected entities $\mathcal{V}$ and their pairwise relations $\mathcal{R}$.

The joint posterior factorizes according to the graph topology:

$$q_\Phi(\mathcal{I}|\mathcal{S}) = \prod_{v_i \in \mathcal{V}} q_\phi(z_i|v_i) \prod_{r_{ij} \in \mathcal{R}} q_\psi(z_{ij}|z_i, z_j) \quad (3)$$

Here, we parameterize these posterior distributions as Gaussians with diagonal covariance $\mathcal{N}(\mu_{ij}, \sigma_{ij})$, where $\sigma_{ij}$ denotes the element-wise standard deviation capturing data-dependent uncertainty in the relational embedding. In this work, we interpret this uncertainty as aleatoric, since it reflects observation-level ambiguity induced by environmental variability rather than model uncertainty. Crucially, the learned $\sigma_{ij}$ provides an input-dependent measure of uncertainty for each entity and interaction, which we use as a proxy for environment-sensitive relations in the subsequent debiasing and pruning steps. Moreover, the variational objective regularizes the posterior through a KL term, discouraging $\sigma_{ij}$ from trivially inflating and encouraging uncertainty to concentrate on hard-to-predict, domain-sensitive relations.

We establish a heuristic connection between uncertainty and spuriousness based on the inherent stochasticity of envi-

ronmental features to guide our disentanglement. Invariant causal interactions generally follow deterministic physical laws, yielding sharp, low-variance representations. In contrast, spurious correlations, such as illumination artifacts, dynamic background clutter or weather textures, often stem from transient environmental contexts. These confounding factors exhibit high intrinsic variance and ambiguity compared to stable physical dynamics.

### 3.3. Uncertainty-Aware Causal Intervention

To disentangle $z_c$ from $z_s$, we introduce a soft feature-space intervention mechanism inspired by the backdoor adjustment criterion. Our objective is to approximate the effect of removing environment-induced spurious correlations by performing a soft, representation-level intervention inspired by the backdoor adjustment principle.

**Environment Space Approximation.** To address the intractability of integrating over the high-dimensional latent environment $\mathcal{E}$ for backdoor adjustment, we employ a non-parametric discretization strategy(Li et al., 2024b). We approximate the continuous environmental space within $\mathcal{E}$ by projecting them onto a finite set of $K$ learnable prototypes, denoted as $\mathcal{C} = \{\mathbf{c}_k\}_{k=1}^{K}$. Each prototype $\mathbf{c}_k \in \mathbb{R}^d$ lies in the same latent space as the relation embedding $z_{ij}$ and captures a recurring environment-dependent context pattern. In this framework, $\mathcal{C}$ functions as a non-parametric support set, serving as a low-rank basis that effectively ap-

proximates the continuous confounding manifold with a tractable, finite support. This enables a tractable approximation of environment-conditioned expectations by a weighted summation over $\mathcal{C}$, $\mathbb{E}_e[\cdot] \approx \sum_{k=1}^{K} P(c_k \mid z_{ij})[\cdot]$.

**Prototype-Based Environment Aggregation.** Subsequently, we leverage this global prior to estimate the environmental context for each relational embedding $z_{ij}$. By querying the prototype set, we infer the alignment probability $P(\mathbf{c}_k|z_{ij})$, which quantifies the propensity of the interaction being governed by the $k$-th environmental mode. Consequently, the continuous environmental expectation $\hat{z}_s^{(ij)}$ is approximated via a propensity-weighted summation:

$$\hat{z}_s^{(ij)} = \sum_{k=1}^{K} P(\mathbf{c}_k|z_{ij}) \cdot \mathbf{c}_k, \quad P(\mathbf{c}_k|z_{ij}) \propto \exp(\frac{\langle z_{ij}, \mathbf{c}_k \rangle}{\sqrt{d}})$$
$$(4)$$

**Uncertainty-Aware Backdoor Adjustment.** Finally, to approximate the environmentally corrected state, we inject this estimated environmental bias $\hat{z}_s^{(ij)}$ into the relational embedding, dynamically modulated by the aleatoric uncertainty $\sigma_{ij}$. Formally, the rectified representation $\tilde{z}_{ij}$ is obtained via a residual correction:

$$\tilde{z}_{ij} = z_{ij} + \phi(\sigma_{ij}) \cdot \hat{z}_s^{(ij)} \qquad (5)$$

where $\phi(\cdot)$ denotes the gating network that maps uncertainty to an intervention intensity. This additive correction can be interpreted as a residual bias compensation mechanism. Since $z_{ij}$ is learned from observations entangled with environment-dependent factors, it may contain a residual bias induced by $z_s$. The aggregated context $\hat{z}_s^{(ij)}$ provides an estimate of this environment-conditioned bias, while the uncertainty-dependent gate $\phi(\sigma_{ij})$ ensures that the correction is applied selectively to unstable relations. As a result, the adjustment attenuates environment-induced confounding effects without overwriting confident, causality-consistent interactions. This constitutes a soft intervention in the representation space, approximating the effect of conditioning on the confounder rather than performing a hard do-operator.

### 3.4. Differentiable Structure Learning and Reasoning

**Differentiable Topology Induction.** To translate the rectified embeddings into a tractable reasoning structure, we perform graph sparsification to induce a sparse topology $\mathcal{S}_{sparse}$. We employ a hybrid pruning strategy that maps each candidate edge to a scalar confidence score and retains only high-confidence relations via adaptive thresholding and per-node Top-$K$ selection. Top-$K$ enforces a sparsity budget so each node keeps its most reliable neigh-

bors, preventing dense connectivity from propagating weak, environment-driven co-occurrences and reducing reasoning cost. In practice, since the score incorporates our adjustment and uncertainty, stable but weak interactions can still rank within the Top-$K$ set.

**Uncertainty-Weighted Reasoning.** We perform reasoning via an Uncertainty-Weighted Message Passing scheme. In the spatial domain, we modulate the Relational GCN (RGCN) aggregation using inverse-variance weights $\tilde{w}_{ij} \propto (\sigma_{ij} + \epsilon)^{-1}$. This mechanism naturally dampens the propagation of unstable, high-variance relations while amplifying robust causal signals. Subsequently, to capture temporal evolution, we aggregate the graph sequence via an LSTM followed by an MLP-based decision head. The final output is modeled as a probabilistic distribution, ensuring that the uncertainty tracked throughout the pipeline is preserved in the final decision.

### 3.5. Optimization Objectives

We formulate a composite objective to jointly optimize prediction accuracy, uncertainty calibration, and structural disentanglement:

$$\mathcal{L} = \mathcal{L}_{\text{pred}} + \lambda_{\text{unc}}\mathcal{L}_{\text{unc}} + \lambda_{\text{div}}\mathcal{L}_{\text{div}} + \lambda_{\text{KL}}\mathcal{L}_{\text{KL}} \qquad (6)$$

where $\mathcal{L}_{\text{pred}}$ is an Aleatoric Cross-Entropy Loss that ensures accurate predictions, particularly for imbalanced classes and critical risk events, using uncertainty-aware loss. $\mathcal{L}_{\text{unc}}$ enforces a ranking-based calibration, penalizing incorrect predictions with lower uncertainty than correct ones. $\mathcal{L}_{\text{div}}$ promotes prototype diversity, fostering robust causal reasoning and preventing overfitting to specific environmental cues. $\mathcal{L}_{\text{KL}}$ regularizes the model's uncertainty distribution by penalizing deviations from the standard normal, preventing over-expansion of uncertainty. Together, these terms guide CURVE toward sparse, stable, and domain-invariant relational reasoning. Detailed formulations of each loss term are provided in Appendix C.

## 4. Experiment

In this section, our experiments are designed to answer three core research questions:

**RQ1 (Generalization):** Does the learned sparse causal structure enable robust generalization to unseen domains?

**RQ2 (Mechanism):** Are the proposed components essential for capturing the true causal topology?

**RQ3 (Reliability):** Is CURVE robust and calibrated for safety-critical tasks?

*Table 1.* Quantitative Analysis of Generalization and Topology. We report in-distribution (ID) results on the CARLA-SR test set and zero-shot OOD performance on DeepAccident. The OOD evaluation is conducted on DeepAccident, which differs from CARLA-SR in both environmental conditions and map topology. Topological complexity is measured by the average node degree and the number of edges. Best and second-best results in each column are highlighted in **bold** and underlined, respectively.

| Method | ID Performance | | | OOD Generalization | | | Topological Complexity | | |
|---|---|---|---|---|---|---|---|---|---|
| | Acc | AUC | MCC | Acc | AUC | MCC | Avg Deg | Avg Edges | Std Edges |
| GCN(Kipf & Welling, 2017) | 89.67 | 96.00 | 62.44 | 62.64 | 67.88 | 28.85 | 73.30 | 545.97 | 417.92 |
| RS2V(Malawade et al., 2022) | **93.36** | 95.81 | **76.86** | 47.25 | 54.40 | -18.03 | **1.90** | **15.77** | **10.36** |
| Sg-risk(Yu et al., 2022) | 88.93 | 95.37 | 62.02 | 56.05 | 56.07 | 11.64 | - | - | - |
| IRM(Arjovsky et al., 2020) | 90.04 | 93.94 | 67.29 | 60.44 | 61.60 | 27.94 | - | - | - |
| DDLoss(Peng et al., 2024) | 89.30 | 95.51 | 63.59 | 63.74 | 66.76 | 34.64 | - | - | - |
| RS2G(Wang et al., 2024a) | 90.77 | 96.07 | 67.61 | 64.84 | 67.35 | **36.44** | 75.92 | 611.25 | 582.85 |
| RS2G (Thr=0.6) | 90.32 | **97.30** | 82.01 | 57.14 | 51.03 | 12.91 | 7.42 | 78.37 | 89.06 |
| CURVE(ours) | 92.92 | 96.19 | 68.85 | **68.13** | **71.15** | 35.07 | 5.78 | 44.80 | 32.14 |

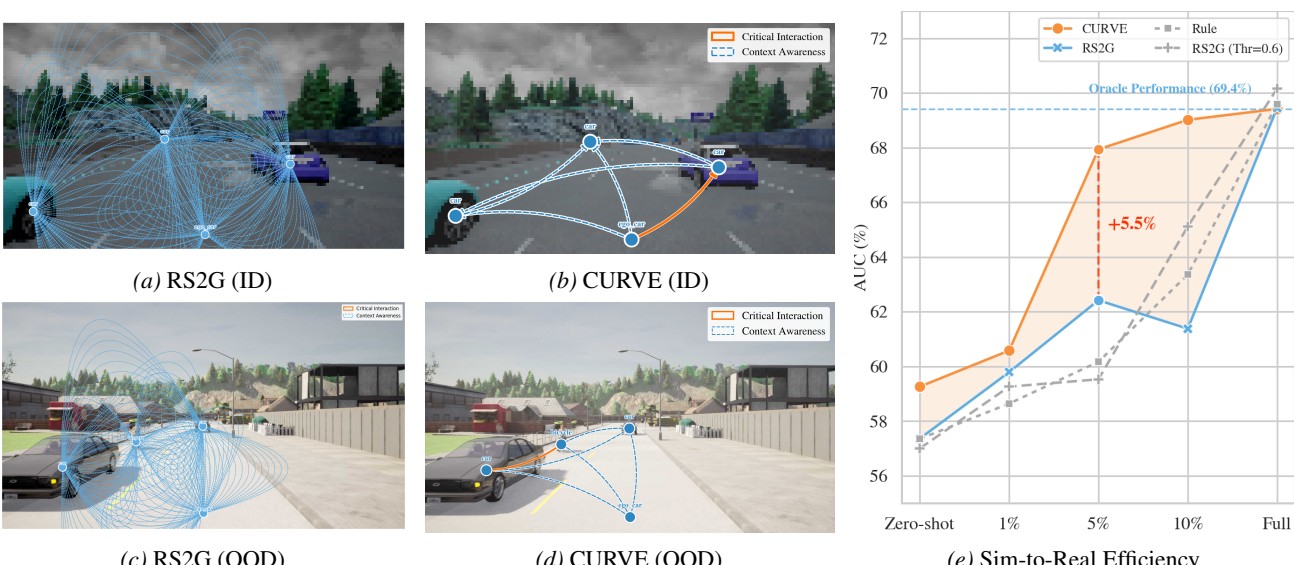

*(a)* RS2G (ID)   *(b)* CURVE (ID)   *(e)* Sim-to-Real Efficiency

*(c)* RS2G (OOD)   *(d)* CURVE (OOD)

*Figure 3.* **Generalization Analysis (RQ1).** (a-d) Structural Sparsity. While the baseline RS2G retains dense connectivity by aggregating diverse redundant correlations between nodes, CURVE utilizes causal intervention to prune spurious edges driven by the environment. (e) Data Efficiency. Adaptation performance (AUC) from simulation to the real world. The results demonstrate the transferability of CURVE, which maintains high accuracy even with minimal annotations.

## 4.1. Experimental Setup

To validate the learned representations, we employ binary collision risk assessment in autonomous driving as the downstream task. Specifically, the task is formulated as a sequence-level binary classification problem. Given a fixed-length image sequence $X$, the model predicts

$$\hat{y} = f_\Phi(X) \in [0, 1], \qquad (7)$$

where $\hat{y} \in \mathbb{R}^2$ denotes the predicted scores for the non-collision and collision classes, respectively. The ground-truth label $Y \in \{0, 1\}$ indicates whether a collision event occurs within the sequence.

For a controlled comparison, we implement CURVE within

the RS2G (Wang et al., 2024a) framework, replacing only topology construction and relation learning with our soft causal intervention and uncertainty-guided pruning, while keeping the backbone, data splits, and training protocol unchanged.

**Datasets.** We validate CURVE across three datasets: (1) CARLA-SR (Dosovitskiy et al., 2017) (Source): A custom synthesized training dataset generated via CARLA Scenario Runner, featuring standardized clear weather and low resolution inputs ($128 \times 72$).

(2) DeepAccident (Wang et al., 2024b) (Zero-shot Simulation): A large scale accident dataset for evaluating generalization. Unlike the source, it contains high resolution

imagery ($1600 \times 900$) with diverse weather conditions and distinct map topologies.

(3) DoTA(Yao et al., 2023) (Simulation to Real World): A real world anomaly dataset used to assess adaptation to physical environments. It introduces significant domain gaps through unstructured roads, complex weather scenarios, and inconsistent camera viewpoints.

**Metrics.** We evaluate performance across three dimensions. For risk assessment performance, we report Accuracy (Acc), Area Under the Curve (AUC), and Matthews Correlation Coefficient (MCC). MCC is specifically chosen for its robustness in evaluating imbalanced binary classifications. For reliability, CURVE outputs the explicit uncertainty to quantify the model's self-confidence in its predictions. To evaluate the sparsity and interpretability, we report Average Degree (Avg Deg) and Average Number of Edges (Avg Edges). Lower values indicate a sparser structure, suggesting the successful removal of spurious correlations. We also report the Standard Deviation of Edge Counts (Std Edges) to measure the stability of the learned causal skeleton across samples.

## 4.2. Generalization Performance

To answer RQ1, we evaluate CURVE across four distinct regimes: in-distribution (ID) evaluation, zero-shot OOD transfer, sim-to-real adaptation, and cross-task transfer to trajectory prediction. The results are summarized in Table 1 and Figure 3.

**Structural Sparsity and Generalization.** Table 1 shows that CURVE achieves a favorable balance between predictive performance and structural sparsity in both ID and zero-shot OOD settings (*CARLA-SR → DeepAccident*). Rule-based methods such as RoadScene2Vec(Malawade et al., 2022) attain sparse graphs through geometric heuristics, but generalize poorly due to their reliance on predefined constraints. In contrast, RS2G(Wang et al., 2024a) relies on dense connectivity when using a low edge confidence threshold, while increasing the threshold to 0.6 yields a sparser graph that overfits to ID data and suffers a sharp degradation in OOD performance. Details of the RS2G thresholding strategy are provided in Appendix D.4. To broaden the comparison beyond graph-specific architectural variants, we additionally include IRM(Arjovsky et al., 2020), a representative OOD generalization objective, and DDLoss(Peng et al., 2024), a recent causality-guided debiasing method, under the same dynamic spatio-temporal backbone and evaluation protocol. CURVE, by comparison, preserves strong OOD accuracy with substantially fewer edges by prioritizing stable, domain-invariant interaction structures over ID-specific shortcuts. This indicates that our pruning mechanism based on uncertainty effectively isolates the stable interaction skeleton by suppressing connections driven by

*Table 2.* **Task transfer on trajectory prediction.** Models are pre-trained on collision risk prediction, with graph backbones frozen during transfer. Lower is better. Entries are reported as FDE/MAE-x/MAE-y.

| Method | ID | OOD |
|---|---|---|
| Rule-based | 18.94/14.75/9.12 | 17.05/12.74/8.80 |
| RS2G | 10.04/8.46/3.71 | 14.70/12.82/3.80 |
| CURVE | 6.71/5.14/3.30 | 9.83/7.68/4.45 |

the environment, rather than relying on dense aggregation (Fig. 3a-3d).

**Adaptation from Simulation to Reality.** As illustrated in Figure 3e, CURVE demonstrates exceptional data efficiency for risk assessment when adapting from simulation to reality (*CARLA-SR → DoTA*). It outperforms baselines in the zero shot setting and achieves parity with the fully supervised oracle using merely 5% of annotations from the real world. This validates that the learned sparse structure is transportable, requiring minimal supervision to bridge the domain gap for accurate hazard prediction.

**Transfer Across Tasks.** To provide additional evidence that CURVE transfers beyond collision risk prediction, we conduct a downstream experiment on trajectory prediction. We freeze the graph backbones of both RS2G and CURVE (pre-trained on collision prediction). We only fine-tune the temporal LSTM alongside a lightweight 2-layer MLP head to predict future waypoints (step 7) based on historical trajectories (steps 1-4). As shown in Table 2, CURVE achieves the best FDE in both ID and OOD settings, with a substantial improvement over RS2G in OOD transfer. This suggests that the learned graph representation remains useful beyond the original collision risk prediction task, providing additional evidence of its generality and transferability under distribution shifts. Therefore, CURVE demonstrates strong potential to serve as a versatile, task-agnostic structural representation for broader autonomous driving perception systems.

## 4.3. Mechanism Verification and Causal Representation

In this section, we answer **RQ2** by verifying whether the proposed components are essential for capturing a more domain-stable and causality-consistent interaction structure from both quantitative and qualitative perspectives.

**Ablation Study.** To answer RQ2, we ablate the prototype-conditioned intervention and uncertainty-guided pruning. Table 3 confirms that both are essential for domain stability. Removing intervention causes a significant OOD MCC drop ($35.07 \rightarrow 25.77$), indicating that without confounder adjustment, the retained edges lack transferability. Conversely, removing uncertainty drastically increases graph density (Avg. Deg. $5.78 \rightarrow 76.61$), verifying its critical role in filtering

*Table 3.* **Ablation Study.** We report Acc/MCC to evaluate performance robustness under class imbalance in ID and OOD scenarios. Additionally, Avg Deg. is included to measure the sparsity of the learned graph topology. Thr= 0.6 and Thr= 0.7 denote threshold-based pruning baselines without uncertainty guidance.

| Variant | ID | OOD | Structure |
|---|---|---|---|
| CURVE (Full) | 92.92/68.85 | 68.13/35.07 | 5.78 |
| w/o Intervention | 89.94/49.93 | 63.74/25.77 | 5.78 |
| w/o Uncertainty | 89.85/49.92 | 65.93/30.46 | 76.61 |
| Thr= 0.6 | 90.32/82.01 | 57.14/12.91 | 7.42 |
| Thr= 0.7 | 86.02/80.06 | 54.86/12.87 | 5.64 |

*Table 4.* Attention weights of representative prototypes across sparse and dense environments.

| Env | $P_{12}$ | $P_0$ | $P_8$ | $P_6$ |
|---|---|---|---|---|
| Sparse | 0.388 | 0.305 | 0.140 | 0.004 |
| Dense | 0.337 | 0.353 | 0.162 | 0.004 |

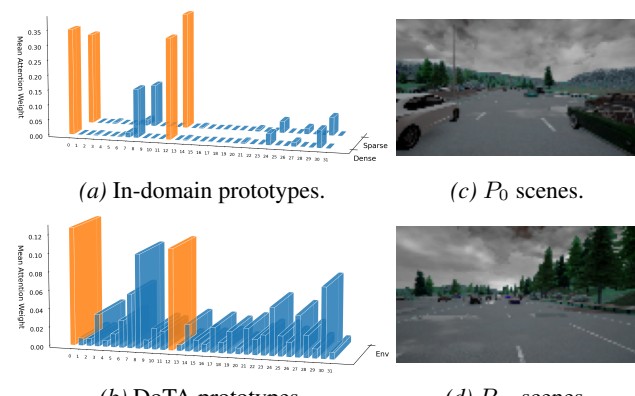

*(a)* In-domain prototypes.    *(c)* $P_0$ scenes.

*(b)* DoTA prototypes.    *(d)* $P_{12}$ scenes.

*Figure 4.* Prototype analysis across domains and traffic contexts. More prototypes become active in real-world DoTA than in the source domain, suggesting reserve capacity under larger domain shifts. Representative scenes also show that different prototypes respond to different traffic contexts.

environment-driven spurious edges. We further compare CURVE with threshold-based pruning strategies derived from RS2G. The original manuscript already includes RS2G (thr= 0.6), which performs direct confidence thresholding without uncertainty guidance. Both threshold-based variants show substantially worse OOD generalization, even at graph sparsity comparable to CURVE, indicating that CURVE benefits not from sparsity alone but from uncertainty-aware pruning of environment-sensitive relations. To make the comparison more controlled, we additionally evaluate a stricter threshold setting, RS2G (thr= 0.7), whose graph density is close to that of CURVE. Additional hyperparameter settings and sensitivity analyses are provided in Appendix D.3.

**Prototype Analysis.** We visualize the learned prototype attention under different traffic contexts in Figure 4 and summarize the quantitative results in Table 4. The attention distribution changes across sparse and dense scenes, indicating that the prototypes capture different latent environmental patterns instead of collapsing to uniform usage. For example, $P_{12}$ is most active in sparse scenes, whereas $P_0$ becomes dominant in dense scenes; $P_8$ plays a secondary role, while $P_6$ remains largely inactive. This suggests that the prototype dictionary learns meaningful context-dependent structure. Moreover, in more complex real-world environments like DoTA dataset, several prototypes become more active than in the source domain, further suggesting that the dictionary retains useful capacity to capture broader environment-dependent variations under larger domain shifts, as shown in Figure 4b.

### 4.4. Reliability and Robustness Analysis

Beyond predictive accuracy, safety-critical scene understanding also requires practical reliability under noisy con-

ditions and efficient inference for online deployment. To address RQ3, we evaluate the reliability of CURVE in safety-critical scenarios from two perspectives: robustness against sensory noise and uncertainty awareness.

**Robustness Analysis.** Real-world environments are inherently noisy. To simulate this, we inject Gaussian noise $\epsilon \sim \mathcal{N}(0, \sigma^2)$ of varying intensities into the input features and measure the performance degradation. As shown in Figure 5a, baseline methods exhibit rapid performance deterioration as noise intensity increases, while CURVE demonstrates superior robustness. This validates that our learned sparse topology effectively acts as a filter, pruning noise-sensitive spurious features and focusing solely on stable causal mechanisms.

**Uncertainty Quantification.** Safety-critical applications demand that a model reflects its own ignorance. We analyze the predictive uncertainty in Figure 5b-5c. We observe a significant distributional shift (large margin between $\mu_{cor}$ and $\mu_{wro}$) where the model assigns high uncertainty to failure cases. Crucially, this separation persists even in the Zero-shot setting, indicating that CURVE remains well-calibrated and capable of distinguishing failures from reliable predictions even under domain shift.

**Computation Overhead and Latency.** To assess the practical overhead of CURVE, we compare its inference efficiency with RS2G on pre-extracted scene-graph sequences, excluding upstream detection for fair comparison. Latency is measured from sequence input to final risk prediction on an RTX 3070 GPU. As shown in Table 5, CURVE introduces only a modest absolute overhead (+0.004M parameters and +6.87 ms latency) while maintaining 76.2 FPS, suggesting that the added uncertainty modeling and prototype-based intervention remain lightweight in practice.

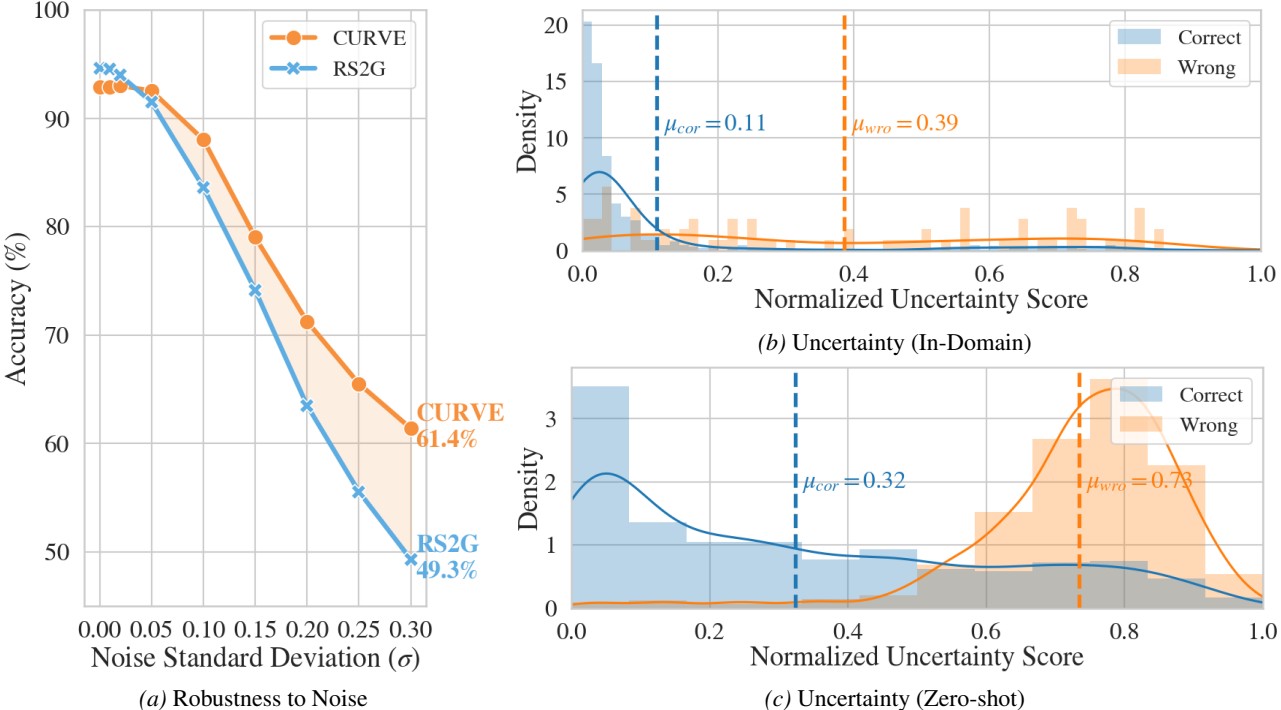

*Figure 5.* **Reliability and Robustness Analysis (RQ3).** (a) Performance degradation under increasing input noise, CURVE remains robust while baselines collapse. (b-c) Uncertainty quantification. The density of uncertainty scores for correct vs. wrong predictions showing clear separation, which indicates the model is well-calibrated even under domain shift.

*Table 5.* **Computational efficiency analysis.** Latency is measured from sequence input to final risk prediction on pre-extracted scene-graph sequences, excluding upstream detection.

| Model | Latency (ms) | FPS | Params (M) | FLOPs (G) |
|---|---|---|---|---|
| RS2G | 6.25 | 159.93 | 0.0191 | 0.0005 |
| CURVE | 13.12 | 76.2 | 0.0231 | 0.0012 |

## 5. Conclusion

In this work, we presented CURVE, a causality-inspired uncertainty-aware representation for vehicle environments. By synergizing variational inference with prototype-driven backdoor adjustment, CURVE explicitly disentangles invariant physical dynamics from environmental biases. Our approach demonstrates superior generalization and robustness across low-data and zero-shot settings, highlighting uncertainty-aware structural learning as a practical pathway toward domain-stable scene understanding in safety-critical environments. While inspired by structural causal modeling, CURVE does not aim to identify true causal graphs or latent confounders; instead, the prototype-based intervention can be viewed as a low-rank, data-driven approximation of environment-dependent biases rather than an explicit causal adjustment. From a practical perspective, CURVE also remains limited by upstream perception quality and may over-prune highly stochastic yet causally relevant interactions.

We discuss these limitations further in Appendix E.

Future work will further investigate the theoretical relationship between uncertainty-guided sparsification and principled causal representation learning, including conditions under which fully causal interpretations may become feasible. In addition, we plan to evaluate the proposed representations in broader end-to-end autonomous driving pipelines, enabling closed-loop testing and application-oriented validation across diverse downstream tasks.

## Acknowledgments

This paper was supported in part by the National Natural Science Foundation of China under Grant 525B2180, in part by the National Natural Science Foundation of China under Grant U23B2061, in part by Shanghai Municipal Science and Technology Major Project under Grant 2021SHZDZX0100, and in part by Xiaomi Young Talents Program/Xiaomi Foundation.

## Impact Statement

This paper presents work whose goal is to advance the field of machine learning. There are many potential societal consequences of our work, none of which we feel must be specifically highlighted here.

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

## A. Notation Table

For clarity, Table 6 summarizes all symbols used throughout this paper.

| Symbol | Description |
| --- | --- |
| $\alpha_{ij}$ | Uncertainty-dependent intervention strength for relation $(i, j)$. |
| $\mathcal{C}$ | Learnable prototype set approximating the latent environment space. |
| $\mathbf{c}_k$ | The $k$-th environmental prototype in $\mathcal{C}$. |
| $d$ | Dimensionality of relational embeddings and prototypes. |
| $e$ | A specific environment instance sampled from $\mathcal{E}$. |
| $\mathcal{E}$ | Environment domain acting as an exogenous variable. |
| $\mathcal{G}$ | Non-linear generative function mapping latent factors to observations. |
| $I(\cdot; \cdot)$ | Mutual information between two random variables. |
| $\mathcal{I}$ | Structured interaction representation produced by $\Phi(x)$. |
| $K$ | Number of environmental prototypes. |
| $\mathcal{L}$ | Overall training objective. |
| $\mathcal{L}_{\mathrm{div}}$ | Prototype diversity regularization loss. |
| $\mathcal{L}_{\mathrm{KL}}$ | KL-divergence regularization for variational uncertainty. |
| $\mathcal{L}_{\mathrm{pred}}$ | Aleatoric prediction loss for risk classification. |
| $\mathcal{L}_{\mathrm{unc}}$ | Uncertainty calibration loss enforcing correct uncertainty ranking. |
| $\mathcal{M}$ | Structural causal model defined as $\langle \mathcal{E}, \mathcal{Z}, \mathcal{G} \rangle$. |
| $\mu_{ij}$ | Mean of the relational posterior distribution. |
| $\tilde{\mu}_{ij}$ | Rectified posterior mean of relation $(i, j)$ after uncertainty-aware causal intervention. |
| $\mathcal{N}(\mu_{ij}, \sigma_{ij})$ | Gaussian posterior with mean $\mu_{ij}$ and diagonal standard deviation $\sigma_{ij}$. |
| $P(\mathbf{c}_k \mid z_{ij})$ | Alignment probability between relation $z_{ij}$ and prototype $\mathbf{c}_k$. |
| $\Phi$ | Scene graph encoder mapping observation $x$ to interaction representation $\mathcal{I}$. |
| $q_\phi(z_i \mid v_i)$ | Posterior distribution of entity latent variable $z_i$. |
| $q_\psi(z_{ij} \mid z_i, z_j)$ | Posterior distribution of relational latent variable $z_{ij}$. |
| $q_\Phi(\mathcal{I} \mid \mathcal{S})$ | Variational posterior of interaction representation conditioned on the scene graph. |
| $r_{ij}$ | Relation between entities $v_i$ and $v_j$. |
| $\mathcal{R}$ | Set of pairwise relations (edges) between entities. |
| $s_{ij}$ | Confidence score of relation $(i, j)$ used for sparsification. |
| $\mathcal{S}$ | Scene graph defined as $\mathcal{S} = (\mathcal{V}, \mathcal{R})$. |
| $\mathcal{S}_{\mathrm{sparse}}$ | Sparse scene graph obtained after differentiable pruning. |
| $\sigma_{ij}$ | Element-wise standard deviation encoding aleatoric uncertainty for relation $z_{ij}$. |
| $\tilde{w}_{ij}$ | Inverse-variance weight used for uncertainty-weighted message passing. |
| $\tilde{z}_{ij}$ | Rectified relational embedding after uncertainty-aware causal intervention. |
| $\mathcal{V}$ | Set of detected entities (nodes) in the scene graph. |
| $v_i$ | The $i$-th entity in the scene graph. |
| $X$ | Raw observation generated by $X = \mathcal{G}(z_c, z_s)$. |
| $Y$ | Downstream prediction target primarily governed by $z_c$. |
| $z_c$ | Invariant causal factors encoding physical scene properties, assumed independent of $e$. |
| $\hat{Z}$ | Learned latent representation inferred from observation $X$. |
| $\mathcal{Z}$ | Set of latent factors, $\mathcal{Z} = \{z_c, z_s\}$. |
| $z_i$ | Latent embedding associated with entity $v_i$. |
| $z_{ij}$ | Latent embedding associated with relation $r_{ij}$. |
| $z_s$ | Spurious environment-dependent factors correlated with $e$. |
| $\hat{z}_s^{(ij)}$ | Estimated environment-conditioned bias for relation $z_{ij}$. |
| $\phi(\cdot)$ | Gating network mapping uncertainty to intervention strength. |
| $\lambda_{\mathrm{div}}, \lambda_{\mathrm{KL}}, \lambda_{\mathrm{unc}}$ | Trade-off coefficients for different loss terms. |

*Table 6.* Summary of symbols used in CURVE.

# B. Mathematical Derivations

This appendix provides a step-by-step mathematical elaboration of the core components of CURVE, following the same processing order as the main framework. The input to the pipeline is a sequence of raw image frames, and the output is a sequence-level prediction together with its associated uncertainty estimate. Specifically, we start from object-level perception and feature construction, then describe variational scene graph generation, uncertainty-aware intervention on relational representations, differentiable structure learning, and finally temporal aggregation and prediction. The loss functions are provided in Appendix C.

### B.1. Object Feature Extraction.

As the first stage of the framework, this module converts raw image observations into structured object-level representations, which are used as the basic entities for subsequent scene graph construction. Given an input image frame $X$, we first extract a set of object instances using an off-the-shelf Mask R-CNN detector (COCO-pretrained, R50-FPN 3x). For each detected instance, the detector outputs a pixel-space bounding box $(x_{\mathrm{L}}, x_{\mathrm{T}}, x_{\mathrm{R}}, x_{\mathrm{B}})$ along with a discrete semantic category label $\tau$. This step operates purely in the observation space and does not impose any relational or graph structure.

To provide a stable geometric reference across frames, we additionally insert fixed anchor instances corresponding to the ego vehicle and lane references. These anchors are assigned predefined bounding boxes in the image coordinate system and are treated identically to detected objects at the feature level.

Each instance is encoded as a fixed-dimensional feature vector $\mathbf{f}_i \in \mathbb{R}^{15}$, defined as

$$\mathbf{f}_i = \big[x_{\mathrm{L}}/W, x_{\mathrm{T}}/H, x_{\mathrm{R}}/W, x_{\mathrm{B}}/H, (x_{\mathrm{R}} - x_{\mathrm{L}})/W, (x_{\mathrm{B}} - x_{\mathrm{T}})/H, \mathbf{o}(\tau)\big], \tag{8}$$

where $W$ and $H$ denote the image width and height. The first six dimensions encode normalized geometric attributes of the bounding box, while $\mathbf{o}(\tau) \in \{0, 1\}^9$ denotes a one-hot semantic encoding over the fixed category set $\{\mathrm{ego\_car}, \mathrm{car}, \mathrm{moto}, \mathrm{bicycle}, \mathrm{ped}, \mathrm{lane}, \mathrm{light}, \mathrm{sign}, \mathrm{road}\}$.

Lane and ego anchors differ only in their assigned geometric attributes and share the same semantic encoding scheme. Across consecutive frames, the resulting set of object feature vectors forms an object feature sequence, which serves as the input to subsequent relational modeling and scene graph construction.

### B.2. Variational Scene Graph Generation

Given the object feature sequence constructed in the previous stage, this module forms a scene-level relational representation by modeling entities and their interactions. Given the object feature sequence $\{\mathbf{f}_i\}_{i=1}^{N}$ extracted from each frame, we construct a probabilistic scene graph representation by defining variational embeddings over entities and their pairwise relations.

**Entity-level latent representation.** Each object instance is treated as an entity and mapped to a latent embedding $z_i$ via a variational encoder. We parameterize a diagonal Gaussian posterior conditioned on the object feature:

$$q_\phi(z_i \mid \mathbf{f}_i) = \mathcal{N}(\mu_i, \mathrm{diag}(\sigma_i^2)), \tag{9}$$

where the mean and log-variance are produced by learnable encoders

$$\mu_i = g_\mu(\mathbf{f}_i), \qquad \log \sigma_i^2 = g_\sigma(\mathbf{f}_i). \tag{10}$$

Sampling is performed via the reparameterization trick,

$$z_i = \mu_i + \sigma_i \odot \epsilon_i, \qquad \epsilon_i \sim \mathcal{N}(0, I), \tag{11}$$

with $\sigma_i = \exp(\frac{1}{2}\log\sigma_i^2)$.

**Relation-level latent representation.** To model interactions between entities, we define a latent relational embedding $z_{ij}$ for each ordered pair $(i, j)$. Conditioned on the corresponding entity embeddings, we parameterize

$$q_\psi(z_{ij} \mid z_i, z_j) = \mathcal{N}(\mu_{ij}, \mathrm{diag}(\sigma_{ij}^2)), \tag{12}$$

where

$$\mu_{ij} = h_\mu([z_i; z_j]), \qquad \log \sigma_{ij}^2 = h_\sigma([z_i; z_j]). \tag{13}$$

Here $[\cdot; \cdot]$ denotes concatenation, and $\sigma_{ij}$ captures data-dependent uncertainty associated with the relation.

### B.3. Uncertainty-Aware Causal Intervention

This module operates on variational relation representations by applying an uncertainty-guided intervention to mitigate environment-dependent relational biases. Specifically, the intervention is applied to the posterior mean $\mu_{ij} \in \mathbb{R}^{|\mathcal{R}|}$ of each relation, prior to sampling the latent variable $z_{ij}$.

**Uncertainty scalar and gate.** Given the relation posterior parameters $\mu_{ij}$ and $\log \sigma_{ij}^2$ (both in $\mathbb{R}^{|\mathcal{R}|}$), we first compute a scalar uncertainty score by averaging the element-wise standard deviation:

$$\sigma_{ij} = \frac{1}{|\mathcal{R}|} \sum_{r=1}^{|\mathcal{R}|} \exp(\tfrac{1}{2} \log \sigma_{ij,r}^2), \tag{14}$$

and map it to an intervention strength $\alpha_{ij} \in (0,1)$ via a learnable gating MLP:

$$\alpha_{ij} = \phi(\sigma_{ij}). \tag{15}$$

Here $\phi(\cdot)$ is implemented as a lightweight gating network consisting of a two-layer MLP with a sigmoid output, ensuring $\alpha_{ij} \in (0,1)$.

**Prototype-based backdoor approximation.** We maintain a learnable prototype dictionary $\mathcal{C} = \{\mathbf{c}_k\}_{k=1}^{K}$, with $\mathbf{c}_k \in \mathbb{R}^{|\mathcal{R}|}$, representing recurring environment-conditioned (confounding) modes. To estimate the environment bias associated with $\mu_{ij}$, we compute attention weights over prototypes using learned query/key projections:

$$\mathbf{q}_{ij} = W_q \mu_{ij}, \qquad \mathbf{k}_k = W_k \mathbf{c}_k, \tag{16}$$

$$a_{ij,k} = \frac{\exp(\mathbf{q}_{ij}^{\top} \mathbf{k}_k)}{\sum_{\ell=1}^{K} \exp(\mathbf{q}_{ij}^{\top} \mathbf{k}_\ell)}, \qquad \hat{z}_s^{(ij)} = \sum_{k=1}^{K} a_{ij,k}\, \mathbf{c}_k. \tag{17}$$

**Residual intervention on the posterior mean.** Finally, we inject the estimated bias $\hat{z}_s^{(ij)}$ into the relation posterior mean, modulated by the uncertainty-dependent gate $\alpha_{ij}$:

$$\tilde{\mu}_{ij} = \mu_{ij} + \alpha_{ij}\, \hat{z}_s^{(ij)}. \tag{18}$$

We denote by $\tilde{z}_{ij}$ the rectified relational embedding sampled from the posterior $\mathcal{N}(\tilde{\mu}_{ij}, \mathrm{diag}(\sigma_{ij}^2))$, which can be expressed via reparameterization as

$$\tilde{z}_{ij} = \tilde{\mu}_{ij} + \sigma_{ij} \odot \epsilon_{ij}, \qquad \epsilon_{ij} \sim \mathcal{N}(0, I). \tag{19}$$

This formulation ensures that the intervention selectively corrects environment-sensitive relations (high $\alpha_{ij}$) while minimally perturbing confident ones (low $\alpha_{ij}$).

### B.4. Differentiable Structure Learning

This module constructs a sparse scene graph topology from rectified relational representations via differentiable confidence scoring and adaptive pruning. Given the rectified relational embeddings $\tilde{\mu}_{ij}$ and their associated uncertainties $\sigma_{ij}$, we induce a sparse graph structure by retaining high-confidence relations through differentiable pruning.

**Relation confidence scoring.** For each candidate relation $(i,j)$, we compute a scalar confidence score

$$s_{ij} = f_{\mathrm{score}}(\tilde{\mu}_{ij}), \tag{20}$$

where $f_{\mathrm{score}}(\cdot)$ is a learnable scoring function implemented as a lightweight MLP.

**Adaptive sparsification.** For each node $i$, we retain only the Top-$K$ relations with the highest scores,

$$\mathcal{N}_i^{(K)} = \mathrm{TopK}\left(\{s_{ij}\}_{j \neq i}, K\right), \tag{21}$$

and define the sparse edge set as

$$\mathcal{R}_{\mathrm{sparse}} = \{(i,j) \mid j \in \mathcal{N}_i^{(K)}\} \cup \{(i,j) \mid s_{ij} > \tau\}, \tag{22}$$

where $\tau$ is a global confidence threshold.

The resulting sparse relation set defines a pruned graph topology $\mathcal{S}_{\text{sparse}}$, which is differentiable with respect to the underlying relation scores and embeddings.

**Uncertainty-weighted message passing.** For each retained relation $(i, j) \in \mathcal{R}_{\text{sparse}}$, messages are weighted by

$$\tilde{w}_{ij} = \frac{1}{\sigma_{ij} + \epsilon}, \tag{23}$$

where $\epsilon$ is a small constant for numerical stability.

While the Top-$K$ operator is non-differentiable, the confidence scores $s_{ij}$ and relational embeddings $\tilde{\mu}_{ij}$ are learned through fully differentiable functions, and gradients are backpropagated through the selected relations.

### B.5. Temporal Aggregation and Prediction Head

This final stage aggregates sparse scene graph representations over time to produce a sequence-level prediction with associated uncertainty. Given a sequence of sparse scene graphs $\{\mathcal{S}_{\text{sparse}}^{(t)}\}_{t=1}^{T}$, we perform spatiotemporal reasoning to obtain a prediction for the entire scene sequence.

**Graph-level representation.** For each time step $t$, we apply relational message passing on the sparse graph $\mathcal{S}_{\text{sparse}}^{(t)}$ to obtain updated node embeddings $\{h_i^{(t)}\}$. A graph-level representation is then obtained via permutation-invariant pooling:

$$g^{(t)} = \text{Pool}\left(\{h_i^{(t)}\}\right), \tag{24}$$

where $\text{Pool}(\cdot)$ denotes global mean pooling in our implementation.

**Temporal modeling.** To capture temporal dependencies across frames, the graph-level representations are aggregated using an LSTM:

$$\mathbf{h}^{(t)} = \text{LSTM}(g^{(t)}, \mathbf{h}^{(t-1)}), \tag{25}$$

where $\mathbf{h}^{(t)}$ denotes the hidden state at time $t$. The final hidden state $\mathbf{h}^{(T)}$ summarizes the temporal evolution of the scene sequence.

**Prediction head.** The sequence-level representation $\mathbf{h}^{(T)}$ is passed to a multilayer perceptron to produce the final prediction:

$$\hat{y} = f_{\text{pred}}(\mathbf{h}^{(T)}), \tag{26}$$

where $f_{\text{pred}}(\cdot)$ denotes the prediction head. Here $\hat{y} \in [0, 1]$ represents the predicted probability of the binary target $Y \in \{0, 1\}$. In addition to the predictive mean, the model outputs an associated data-dependent uncertainty, which is propagated from relational embeddings and used to quantify confidence in the prediction.

## C. Optimization Objectives

We optimize CURVE with a composite objective that combines aleatoric risk prediction, uncertainty calibration, prototype diversity regularization, and variational KL regularization. The final training loss is

$$\mathcal{L} = \mathcal{L}_{\text{pred}} + \lambda_{\text{unc}}\mathcal{L}_{\text{unc}} + \lambda_{\text{div}}\mathcal{L}_{\text{div}} + \lambda_{\text{KL}}\mathcal{L}_{\text{KL}} \tag{27}$$

Unless otherwise stated, we set the loss weights to $\lambda_{\text{unc}} = 1.0$, $\lambda_{\text{div}} = 10^{-2}$, and $\lambda_{\text{KL}} = 10^{-3}$.

### C.1. Aleatoric Prediction Loss $\mathcal{L}_{\text{pred}}$

The prediction head outputs class-wise logits together with a data-dependent uncertainty, parameterized by a mean $\text{logits}_\mu \in \mathbb{R}^2$ and a log-variance $\text{logits}_{\log \sigma^2} \in \mathbb{R}^2$. The corresponding standard deviation is obtained as

$$\text{logits}_\sigma = \exp\left(\tfrac{1}{2}\text{logits}_{\log \sigma^2}\right). \tag{28}$$

During training, we inject Gaussian noise into the logits to account for aleatoric uncertainty,

$$\text{logits}^{(m)} = \text{logits}_\mu + \epsilon^{(m)} \odot \text{logits}_\sigma, \qquad \epsilon^{(m)} \sim \mathcal{N}(0, I), \tag{29}$$

and compute the prediction loss by averaging cross-entropy over $M$ Monte Carlo samples,

$$\mathcal{L}_{\text{pred}} = \frac{1}{M} \sum_{m=1}^{M} \text{CE}(\text{logits}^{(m)}, Y), \tag{30}$$

where $\text{CE}(\cdot)$ denotes cross entropy for the binary target $Y \in \{0, 1\}$. We use $M = 10$ in all experiments.

### C.2. Uncertainty Calibration Loss $\mathcal{L}_{\text{unc}}$

We encourage the model to assign higher predictive uncertainty to misclassified samples than to correct ones. Predicted labels are obtained from the mean logits,

$$\hat{y}_n = \arg\max_c \text{logits}_{\mu, n, c}. \tag{31}$$

For samples predicted correctly and incorrectly within a mini-batch, we compute the average predicted standard deviation

$$u_{\text{cor}} = \mathbb{E}[\exp(\tfrac{1}{2}\text{logits}_{\log \sigma^2}) \mid \hat{y} = y], \quad u_{\text{wro}} = \mathbb{E}[\exp(\tfrac{1}{2}\text{logits}_{\log \sigma^2}) \mid \hat{y} \neq y]. \tag{32}$$

We enforce uncertainty separation using a hinge ranking loss with margin $\delta = 0.1$,

$$\mathcal{L}_{\text{rank}} = \max(0, \ u_{\text{cor}} - u_{\text{wro}} + \delta), \tag{33}$$

and prevent uncertainty collapse on incorrect predictions via a minimum threshold $u_{\text{min}} = 0.5$,

$$\mathcal{L}_{\text{collapse}} = \max(0, \ u_{\text{min}} - u_{\text{wro}}). \tag{34}$$

We define $\mathcal{L}_{\text{unc}} = \mathcal{L}_{\text{rank}} + \mathcal{L}_{\text{collapse}}$.

### C.3. Prototype Diversity Loss $\mathcal{L}_{\text{div}}$

To prevent the learned prototype dictionary from collapsing to redundant directions, we explicitly encourage diversity among prototype vectors. Let $\mathcal{C} = \{\mathbf{c}_k\}_{k=1}^{K}$ denote the prototype dictionary. We first normalize each prototype to unit length,

$$\tilde{\mathbf{c}}_k = \frac{\mathbf{c}_k}{\|\mathbf{c}_k\|_2}, \tag{35}$$

and compute the pairwise cosine similarity matrix

$$S_{ij} = \tilde{\mathbf{c}}_i^\top \tilde{\mathbf{c}}_j. \tag{36}$$

To encourage orthogonality between different prototypes, we penalize deviations from the identity matrix using a Frobenius norm:

$$\mathcal{L}_{\text{div}} = \|S - I\|_{\text{F}}. \tag{37}$$

This loss encourages low similarity between distinct prototypes while preserving self-similarity on the diagonal. In all experiments, we use a fixed prototype dictionary size of $K = 32$.

## C.4. KL Regularization $\mathcal{L}_{\text{KL}}$

To prevent the predicted uncertainty of nodes and relations from arbitrarily inflating, we regularize the variance parameters of the variational posteriors using a KL-divergence penalty. Specifically, for the node- and relation-level posteriors $q_\phi(z_i \mid \cdot)$ and $q_\psi(z_{ij} \mid \cdot)$, parameterized by a diagonal Gaussian $\mathcal{N}(\mu, \text{diag}(\sigma^2))$, we penalize deviations of the variance $\sigma^2$ (and the corresponding mean $\mu$) from a unit Gaussian prior:

$$\text{KL}(\mathcal{N}(\mu, \text{diag}(\sigma^2)) \,\|\, \mathcal{N}(0, I)) = \frac{1}{2} \sum_\ell (\mu_\ell^2 + \sigma_\ell^2 - \log \sigma_\ell^2 - 1). \tag{38}$$

The total KL loss $\mathcal{L}_{\text{KL}}$ sums the above terms over all node- and relation-level posteriors.

# D. Experimental Details

This section details the experimental setup used to evaluate CURVE. We first describe the datasets and their roles in modeling in-distribution learning, zero-shot out-of-distribution generalization, and sim-to-real transfer. We then introduce the training and evaluation regimes under different supervision levels, followed by a summary of hyperparameter settings and baseline implementation details to ensure fair comparison. Finally, we present the sim-to-real adaptation protocol, oracle definition, and robustness evaluation under controlled noise perturbations, together with detailed numerical results that complement the figures reported in the main paper.

## D.1. Datasets

We evaluate our method on three datasets that represent complementary generalization regimes in autonomous driving, including a synthetic in-distribution source dataset, a zero-shot simulation-based OOD benchmark, and a real-world sim-to-real adaptation dataset. Together, they enable a systematic assessment of in-distribution performance, cross-domain generalization, and data-efficient transfer. Table 7 summarizes the key statistics of the datasets used in our experiments, including the number of data sequences and the class distribution for risk assessment.

*Table 7.* Dataset statistics for experimental evaluation.

| Dataset | Domain | Number of Sequences | Number of Frames | Positive Ratio | Resolution |
|---|---|---|---|---|---|
| CARLA-SR | Simulation (ID) | 1043 | 10437 | 14.00% | $128 \times 72$ |
| DeepAccident | Simulation (OOD) | 91 | 7260 | 45.05% | $1600 \times 900$ |
| DoTA | Real World | 4614 | 46140 | 45.92% | $128 \times 72$ |

CARLA-SR is generated using the CARLA Scenario Runner with standardized map and weather configurations and serves as the source domain for training, consistent with RoadScene2Vec and RS2G. The dataset is relatively simple, focusing primarily on lane-changing and related traffic scenarios, with limited scene diversity and low image resolution.

DeepAccident is designed to introduce substantial domain shifts and is used exclusively for zero-shot OOD evaluation. Compared to CARLA-SR, it exhibits much greater diversity in scene layout, object density, camera viewpoints, and visual appearance, with significantly higher image resolution and more complex, accident-prone interactions. The dataset spans 7 CARLA town maps (Town01, 02, 03, 04, 05, 07, and 10) and covers 14 weather conditions, including ClearNoon, ClearNight, ClearSunset, CloudyNoon, CloudySunset, SoftRainNoon, SoftRainSunset, MidRainNoon, MidRainSunset, HardRainNoon, HardRainSunset, HardRainNight, WetCloudyNoon, and WetCloudySunset. Since the original dataset does not provide sequence-level labels, we manually annotate sequence-level collision labels for our experiments.

DoTA consists of real-world driving videos and is used to evaluate sim-to-real generalization under limited annotation budgets. It is constructed from over 6,000 YouTube video clips, from which diverse dash-camera accident videos were selected across different countries, weather conditions, and lighting environments. Videos with unobservable accidents or camera detachment are excluded, and the remaining videos are sampled at 10 fps for annotation and experimentation in this work.

## D.2. Training and Evaluation Regimes

All methods are evaluated under three complementary regimes: in-distribution (ID) learning, zero-shot out-of-distribution (OOD) generalization, and sim-to-real transfer under varying levels of target-domain supervision.

**In-Distribution (ID).** For ID evaluation, models are trained, validated, and tested exclusively on the *CARLA-SR* dataset using fixed splits (training/validation/testing = 7/1/2).

**Zero-Shot Out-of-Distribution Generalization (*CARLA-SR → DeepAccident*).** For zero-shot evaluation, models are trained only on CARLA-SR and directly evaluated on the DeepAccident dataset without accessing any target-domain labels. No fine-tuning, hyperparameter re-selection, or threshold adjustment is performed on the target domain, and all model parameters remain frozen during evaluation.

**Sim-to-Real Transfer with Partial Supervision (*CARLA-SR → DoTA*).** To evaluate sim-to-real transfer under increasing levels of target-domain supervision, we consider a range of supervision ratios, including 0%, 1%, 5%, 10%, and 100%. The 0% setting corresponds to zero-shot sim-to-real transfer, where the model trained on CARLA-SR is directly evaluated on DoTA without adaptation. For low-data adaptation (1%, 5%, and 10%), models are initialized from the CARLA-SR-trained checkpoint. To ensure a consistent and fair adaptation protocol, we freeze the graph construction and representation learning modules, and fine-tune only the temporal module (LSTM) and the prediction head for 50 epochs. The 100% supervision setting corresponds to a fully supervised oracle, in which all model parameters are unfrozen and optimized on the full DoTA training set for 200 epochs. This setting serves as an upper bound on performance when complete target-domain supervision is available.

## D.3. Hyperparameter Settings and Sensitivity Analysis

Table summarizes the key hyperparameter settings used in our experiments, together with additional sensitivity analyses on the prototype size $K$, prototype dimension $d$, and uncertainty loss weight $\lambda_{unc}$. For fair comparison, all methods share the same training configuration whenever applicable. CURVE follows the RS2G backbone settings, while its additional hyperparameters are kept fixed unless explicitly varied in the ablations.

Overall, the results show that CURVE is relatively stable to the prototype size and prototype dimension. Varying $K$ or $d$ leads to only modest changes in OOD performance and graph complexity, suggesting that the gains of CURVE do not rely on brittle hyperparameter choices. In contrast, the uncertainty loss weight $\lambda_{unc}$ has a clearer influence: overly small values (e.g., $0.1$ or $0.5$) noticeably degrade OOD performance, while $1.0$ and $2.0$ give similar results. Based on the overall ID/OOD trade-off, we use $K = 32$, $d = 64$, and $\lambda_{unc} = 1.0$ as the default configuration in all experiments.

## D.4. Baseline Clarification

All baseline methods are implemented following their original papers or official codebases. For graph-based methods, we use identical node definitions, input features, and data splits across all experiments. Importantly, CURVE is implemented within the RS2G framework and differs only in topology construction and relation learning. All other components, including the backbone architecture, temporal module, and training protocol, are kept identical to ensure a fair comparison.

Specially, RS2G constructs scene graphs by assigning a confidence score to each candidate edge and retaining edges whose scores exceed a predefined threshold. In the original implementation, a relatively low threshold (0.25) is adopted, such that edges with even weak confidence are preserved, resulting in densely connected graphs. To examine the effect of confidence-based graph sparsification, we additionally evaluate RS2G with a higher edge confidence threshold of 0.6, which is slightly above random chance and retains only high-confidence relations. This threshold is selected based on validation performance on the source domain and is fixed across all evaluation regimes. This setting follows the same scoring mechanism as RS2G and only modifies the acceptance threshold.

## D.5. Sim-to-Real Adaptation and Oracle Definition

In the sim-to-real adaptation experiments, the oracle performance refers to RS2G trained with full supervision on the target dataset. Specifically, the oracle model is trained using 100% labeled data from DoTA, following the same architecture and training protocol as other adaptation settings, with all model parameters unfrozen. Table 9 reports detailed sim-to-real adaptation results across different supervision levels, complementing Figure 3e in the main paper.

*Table 8.* Key hyperparameter settings and sensitivity analysis of CURVE. The left panel reports the shared experimental configuration, and the right panels show ablations on prototype size $K$, prototype dimension $d$, and uncertainty loss weight $\lambda_{unc}$.

**(a) Key hyperparameters**

| Hyperparameter | Value |
|---|---|
| Task type | Sequence classification |
| Optimizer | Adam |
| Learning rate | $1 \times 10^{-4}$ |
| Epochs | 500 |
| Batch size | 64 |
| Weight decay | 0.1 |
| Data split | train:val:test = 7:1:2 |
| Dropout | 0.5 |
| GNN conv | FastRGCNConv |
| Hidden dimension | 64 |
| Pooling | SAGPool (ratio 0.5) |
| Readout | Mean |
| Temporal module | LSTM-attn ($50 \rightarrow 20$) |
| Output classes | 2 |

**(b) Ablation on prototype size $K$**

| Variant | ID(Acc/MCC) | OOD(Acc/MCC) | AvgDeg. |
|---|---|---|---|
| $K = 8$ | 90.61/55.98 | 67.03/32.78 | 5.75 |
| $K = 16$ | 91.28/60.46 | 68.13/35.62 | 5.76 |
| $K = 32$ (Def.) | 92.92/68.85 | 68.13/35.07 | 5.78 |
| $K = 64$ | 91.28/59.82 | 67.03/33.04 | 5.57 |

**(c) Ablation on prototype dimension $d$**

| Variant | ID(Acc/MCC) | OOD(Acc/MCC) | AvgDeg. |
|---|---|---|---|
| $d = 32$ | 91.47/66.78 | 67.03/32.83 | 5.85 |
| $d = 64$ (Def.) | 92.92/68.85 | 68.13/35.07 | 5.78 |
| $d = 128$ | 91.18/60.26 | 67.03/32.77 | 5.58 |

**(d) Ablation on uncertainty loss weight $\lambda_{unc}$**

| Variant | ID(Acc/MCC) | OOD(Acc/MCC) | AvgDeg. |
|---|---|---|---|
| $\lambda_{unc} = 0.1$ | 88.98/64.83 | 60.43/22.93 | 5.80 |
| $\lambda_{unc} = 0.5$ | 91.09/59.05 | 65.93/30.73 | 5.85 |
| $\lambda_{unc} = 1.0$ (Def.) | 92.92/68.85 | 68.13/35.07 | 5.78 |
| $\lambda_{unc} = 2.0$ | 90.61/59.44 | 68.13/36.02 | 5.77 |

*Table 9.* Sim-to-real adaptation performance on DoTA under different supervision levels. Results are reported as Acc/AUC/MCC. The full supervision setting corresponds to the oracle performance.

| Method | Zero-shot | 1% | 5% | 10% | Full (Oracle) |
|---|---|---|---|---|---|
| Rule | 55.07/57.35/6.30 | 55.98/58.64/12.43 | 57.87/60.17/13.68 | 59.45/63.36/17.24 | 64.43/69.60/29.16 |
| RS2G | 56.76/57.36/13.11 | 58.28/59.79/15.50 | 58.17/62.42/15.11 | 59.17/61.38/16.78 | 64.35/69.46/28.66 |
| RS2G (Thr=0.6) | 55.98/57.00/11.48 | 56.76/59.26/12.07 | 57.24/59.53/12.28 | 62.68/65.12/24.17 | 65.08/70.16/29.16 |
| CURVE | **58.50/59.26/15.55** | **58.80/60.58/15.65** | **63.72/67.94/27.47** | **64.35/69.02/27.91** | **65.50/69.42/30.33** |

## D.6. Robustness Evaluation and Noise Injection

Robustness is evaluated by injecting Gaussian noise into the geometric components of the object feature vectors defined in Eq. 8. Specifically, noise is applied to the normalized bounding box coordinates $[x_L/W, x_T/H, x_R/W, x_B/H]$, while all semantic components remain unchanged. Formally, given the geometric subvector $\mathbf{x} \in [0,1]^4$, we apply:

$$\tilde{\mathbf{x}} = \text{clip}(\mathbf{x} + \epsilon, 0, 1), \quad \epsilon \sim \mathcal{N}(0, \sigma^2). \tag{39}$$

The perturbed coordinates are then used to compute derived geometric attributes such as width and height, following the original feature construction pipeline. The one-hot semantic encoding $\mathbf{o}(\tau)$ is not perturbed. This perturbation is applied only during evaluation and is shared across all compared methods. By operating directly on object-level geometric features, this protocol isolates the robustness of the learned graph structure and relational reasoning from variations in raw image appearance or detector performance. Table 10 reports detailed robustness results across different noise levels, complementing Figure 5a in the main paper.

## D.7. Validation of Causal-Invariant Dynamics

To verify the intrinsic causal mechanism, we visualize the latent space of the learned invariant factors ($z_c$) in Figure 6. Ideally, robust causal representations for dynamic scenes should capture the continuous physical process of risk accumulation rather than treating frames as independent static samples. Consistent with this, our representation reveals a continuous risk evolution manifold. This structure confirms that CURVE captures the continuous physical process of risk accumulation rather than treating frames as independent static samples. By encoding the temporal progression of risk, the model successfully

*Table 10.* Robustness performance under different Gaussian noise intensities (reported in %).

| $\sigma$ | CURVE | | | RS2G | | |
|---|---|---|---|---|---|---|
| | Acc | AUC | MCC | Acc | AUC | MCC |
| 0.00 | 92.91 | 96.21 | 68.82 | 94.64 | 98.57 | 78.83 |
| 0.01 | 92.91 | 96.34 | 68.85 | 94.54 | 98.49 | 78.37 |
| 0.02 | 93.01 | 96.34 | 68.97 | 93.97 | 98.29 | 75.75 |
| 0.05 | 92.53 | 96.40 | 67.60 | 91.48 | 97.37 | 68.58 |
| 0.10 | 88.03 | 93.13 | 54.93 | 83.62 | 94.14 | 51.98 |
| 0.15 | 79.02 | 82.62 | 41.72 | 74.14 | 89.01 | 40.04 |
| 0.20 | 71.26 | 73.64 | 32.86 | 63.51 | 80.68 | 33.22 |
| 0.25 | 65.52 | 65.24 | 28.02 | 55.56 | 73.13 | 30.22 |
| 0.30 | 61.40 | 60.30 | 25.96 | 49.33 | 65.49 | 26.68 |

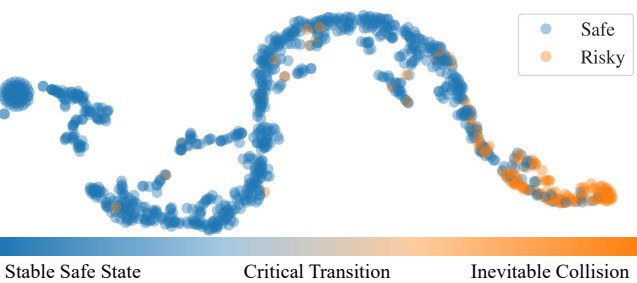

*Figure 6.* Visualization of the learned features. The t-SNE projection reveals a risk evolution manifold that reflects the temporal dynamics of accidents. The trajectory evolves from concentrated stable safe states, through a critical transition phase where risk accumulates, to inevitable collision states, demonstrating the model's capacity to encode physical progression.

disentangles invariant physical dynamics from spurious environmental correlations, ensuring robust decision-making across diverse domains.

## E. Scope and Limitations

We discuss the scope of applicability and limitations of CURVE to clarify its intended use cases and underlying assumptions.

**Causal Interpretation Boundary.** CURVE is causality-inspired rather than a causal discovery or identification framework. While we adopt structural causal modeling as a conceptual lens, the proposed prototype-conditioned intervention does not guarantee recovery of true structural causal graphs or latent causal factors. In particular, the learned prototypes serve as a low-rank approximation of recurring environment-dependent modes and should not be interpreted as explicit or identifiable confounders. Our objective is not causal identifiability, but to induce domain-stable interaction structures that empirically improve out-of-distribution generalization.

**Task and Application Scope.** Our experimental validation focuses on sequence-level collision risk assessment under distribution shifts. Although the learned sparse relational representations are promising for robust scene understanding, CURVE is not evaluated in closed-loop planning, control, or trajectory optimization settings. Extending the framework to downstream decision-making and policy learning remains future work.

**Failure Cases.** Despite its robustness under distribution shift, CURVE may fail in two representative scenarios.

First, CURVE is fundamentally limited by upstream perception quality. Similar to most graph-based scene understanding methods, our framework operates on object-level scene graphs constructed from detected instances. If a safety-critical object is missed by the upstream detector, for example due to severe occlusion, truncation, motion blur, or extremely poor visibility, no corresponding node can be instantiated in the graph. In such cases, the downstream causal intervention and uncertainty-guided sparsification cannot recover the missing evidence, since the relevant interaction is absent from the graph from the outset. This limitation is inherited from the perception front-end rather than from the proposed causal regularization itself.

Second, CURVE may over-prune relations that are both causally important and intrinsically high-variance. Our sparsification

heuristic is motivated by the empirical observation that highly uncertain relations are often associated with environment-sensitive noise or unstable spurious correlations. While this assumption is effective in most cases, it is not universally valid. In particular, some truly safety-critical interactions are inherently difficult to predict, such as an erratic pedestrian, an abruptly merging vehicle, or a rare near-collision maneuver. These relations may legitimately exhibit high uncertainty even though they are causally relevant. In such cases, the model may mistakenly suppress or underweight an important connection, which can lead to missed risk cues or delayed recognition of hazardous situations.

These failure cases highlight that uncertainty should be interpreted as a useful but imperfect proxy for spuriousness. Future work could address these limitations by jointly improving upstream perception robustness and developing more selective sparsification mechanisms that better distinguish unpredictable-but-causal relations from purely environment-driven noise.

**Summary.** Despite these limitations, CURVE provides a practical and effective framework for learning domain-stable relational representations by leveraging uncertainty-guided structural regularization. We believe this work represents a meaningful step toward robust scene understanding under distribution shifts, while leaving theoretical identifiability and closed-loop integration as important directions for future research.

