# OpenReview forum: "CURVE: Learning Causality-Inspired Invariant Representations for Robust Scene Understanding via Uncertainty-Guided Regularization"
_ICML.cc/2026/Conference — ICML 2026 regular_

### Official Review · Reviewer_TLey · 2026-03-10

**Soundness:** 3
**Presentation:** 3
**Significance:** 3
**Originality:** 3
**Overall Recommendation:** 4
**Confidence:** 2

**Summary:**

This paper proposes CURVE, a framework for learning robust scene graph representations under distribution shifts in autonomous driving environments. The method combines variational uncertainty modeling, prototype-based confounder approximation, and uncertainty-guided graph sparsification to suppress spurious environment-dependent relations.

The key idea is to treat relational uncertainty as a signal of environment-induced spurious correlations. The model performs a soft intervention in feature space using learnable prototypes that approximate environmental confounders and applies differentiable pruning to produce sparse scene graph topologies. The resulting graph representations are then used for downstream risk prediction.

Experiments are conducted on CARLA-SR (source), DeepAccident (zero-shot OOD), and DoTA (sim-to-real) datasets for collision risk prediction. Results show improved OOD performance and sparser graph structures compared to baselines such as RS2G.

**Compliance With Llm Reviewing Policy:**

Affirmed.

**Final Justification:**

The authors' rebuttal addressed my main concerns by adding causal/OOD baseline comparisons (showing CURVE outperforms IRM and DDLoss), providing an exact ablation (demonstrating that uncertainty pruning alone underperforms the full model at comparable sparsity), adding a trajectory prediction experiment (showing generalization beyond collision risk prediction), and committing to formatting fixes. I therefore maintain my Weak Accept recommendation.

**Key Questions For Authors:**

Please see the weaknesses.

**Limitations:**

yes

**Strengths And Weaknesses:**

Strengths:

1. The paper clearly identifies a real issue: dense scene graph connectivity often captures environment-specific correlations rather than invariant interaction dynamics. This is a well-known challenge for OOD generalization in driving perception systems.

2. The learnable prototype dictionary used to approximate environmental confounders is a reasonable approximation to backdoor adjustment in high-dimensional latent spaces. The attention-based aggregation mechanism is technically simple and easy to implement.

3. The paper attempts to evaluate structural sparsity (avg degree, edges), which is valuable when proposing methods for structure learning in graphs.

Weaknesses:

1. The baselines mainly consist of scene graph models and variations of the RS2G framework. However, the core claim of the paper concerns robustness under distribution shift. It would therefore be useful to compare against methods from the domain generalization or invariant learning literature (e.g., approaches inspired by invariant risk minimization or causal representation learning).

2. The proposed approach includes several interacting modules: variational graph generation, prototype dictionaries, uncertainty gating, sparsification, and multiple regularization losses. Although the ablation study partially addresses this, it still remains somewhat unclear how necessary each component is. In particular, it would be interesting to know whether simpler approaches (e.g., uncertainty-based edge pruning without the prototype intervention) could achieve similar gains.

3. All experiments focus on a single downstream application, namely binary collision risk prediction. While this is an important task, it is difficult to assess how general the learned representation is. Additional experiments on other scene understanding tasks (e.g., trajectory prediction or scene graph reasoning) would strengthen the empirical evidence.

4. The formatting of equations is somewhat inconsistent. Some equations are followed by punctuation marks (e.g., periods or commas), while others are not. In addition, the font size used in several figures appears relatively large; adjusting it slightly may improve the overall visual consistency and readability of the paper.

---

> ### Author Rebuttal · Authors · 2026-03-28
>
> We thank Reviewer TLey for the positive and constructive review. We appreciate your recognition of the motivation, prototype-based design, and structural sparsity analysis in our work. Below, we address your specific concerns.
>
> ### 1. Comparisons with Causal/OOD Baselines (W1).
> To better evaluate robustness under distribution shift, we add comparisons between CURVE and representative methods from invariant and causal learning, including **IRM**, a widely used objective for OOD generalization, and **DDLoss**, a causality-guided debiasing method. For a fair comparison, we implement their core objectives on the same backbone, while keeping the architecture and training protocol unchanged. As shown in Table I, CURVE achieves the best performance in both ID and OOD settings.
>
> Table I. Comparisons with Causal/OOD Baselines.
> |Method|ID (Acc/AUC/MCC)|OOD (Acc/AUC/MCC)|
> |:---|:---:|:---:|
> |IRM|90.04/93.94/67.29|60.44/61.60/27.94|
> |DDLoss|89.30/95.51/63.59|63.74/66.76/34.64|
> |CURVE (ours)|92.92/96.19/68.85|68.13/71.15/35.07|
>
> ### 2. Component Necessity and Pruning Ablations (W2).
> To directly address the reviewer's question of whether similar gains can be achieved by a simpler design, we compare CURVE with several pruning-based variants. In particular, the reviewer's suggested setting of **uncertainty-based edge pruning without prototype intervention** corresponds to our **w/o Intervention** variant. We further include **RS2G (Thr=0.6)** and a new **RS2G (Thr=0.7)** baseline, where the latter closely matches CURVE's sparsity level.
>
> Table II. Ablations of Pruning Strategies.
> |Method|ID(Acc/MCC)|OOD(Acc/MCC)|Complexity(Avg Deg.)|
> |:---|:---:|:---:|:---:|
> |Thresholding-Pruning (Thr=0.6)|90.32/82.01|57.14/12.91|7.42|
> |Thresholding-Pruning (Thr=0.7)|86.02/80.06|54.86/12.87|5.64|
> |Uncertainty-Based Pruning w/o Intervention|89.94/49.93|63.74/25.77|5.64|
> |CURVE (ours)|92.92/68.85|68.13/35.07|5.78|
>
> As shown in Table II, uncertainty-guided pruning alone improves OOD performance over threshold-based baselines, but still underperforms the full model at comparable graph complexity. At nearly matched average degree, CURVE achieves better OOD performance (**68.13/35.07** vs. 63.74/25.77 in Acc/MCC), supporting the benefit of the prototype-based intervention beyond pruning alone.
>
> Regarding the other interacting modules you noted, they are designed to work together. In particular, the variational graph generation provides the relational variance estimates used by the uncertainty gating. To further examine the role of uncertainty regularization, we ablate the uncertainty loss weight $\lambda_{unc}$ in Table III.
>
> Table III. Ablation on Uncertainty Loss Weight ($\lambda_{unc}$).
> |Variant|ID (Acc/MCC)|OOD (Acc/MCC)|Complexity (Avg Deg.)|
> |:---|:---:|:---:|:---:|
> |$\lambda_{unc}=0.1$|88.98/64.83|60.43/22.93|5.80|
> |$\lambda_{unc}=0.5$|91.09/59.05|65.93/30.73|5.85|
> |$\lambda_{unc}=1.0$ (Def.)|92.92/68.85|68.13/35.07|5.78|
> |$\lambda_{unc}=2.0$|90.61/59.44|68.13/36.02|5.77|
>
> The results show that overly small regularization weights (e.g., 0.1) noticeably degrade OOD performance, while the performance remains relatively stable within a reasonable range. This suggests that uncertainty regularization is important, but the model is **not overly sensitive** to the exact choice of $\lambda_{unc}$.
>
> ### 3. Evaluation on Additional Downstream Task (W3).
> To assess whether the learned representation transfers beyond binary collision risk prediction, we conduct an additional downstream experiment on **trajectory prediction**. Specifically, we freeze the graph backbone and fine-tune only the LSTM together with a lightweight 2-layer MLP prediction head. In this experiment, the scene graph construction remains unchanged, while the downstream supervision is replaced from binary risk labels to future spatial coordinates. The task is to predict the future waypoint (step 7) from historical trajectories (steps 1–4). The rule-based baseline assumes constant velocity.
>
> Table IV. Trajectory Prediction Results.
> |Method|ID (FDE/MAE-x/MAE-y)|OOD (FDE/MAE-x/MAE-y)|
> |:---|:---:|:---:|
> |Rule-based|18.94/14.75/9.12|17.05/12.74/8.80|
> |RS2G|10.04/8.46/3.71|14.70/12.82/3.80|
> |CURVE|6.71/5.14/3.30|9.83/7.68/4.45|
>
> As shown in Table IV, CURVE achieves the best FDE in both ID and OOD settings, with a substantial improvement over RS2G in OOD transfer. This suggests that the learned graph representation remains useful beyond the original collision risk prediction task, providing additional evidence of its **generality and transferability** under distribution shifts.
>
> ### 4. Presentation and Formatting (W4).
> We appreciate the reviewer's careful reading and apologize for the presentation inconsistencies. In the final version, we will standardize equation punctuation throughout the paper and reduce the figure font sizes to improve overall consistency and readability.

---

> > ### Author Rebuttal · Reviewer_TLey · 2026-04-03
> >
> > Thank you for the thorough and well-structured rebuttal. The additional experiments significantly strengthen the paper. In particular, the comparisons with causal/OOD baselines, the pruning ablations addressing simpler alternatives, and the added trajectory prediction task effectively address my main concerns regarding evaluation and component necessity. Overall, the rebuttal improves confidence in both the method and empirical validation.

---

> > > ### Author Response · Authors · 2026-04-03
> > >
> > > Thank you very much for your thoughtful and encouraging feedback! We are very happy to know that our clarifications and additional experiments have addressed your concerns. Your helpful comments and suggestions have also helped us improve the quality and clarity of the paper. We sincerely appreciate your time and careful evaluation.

---

### Official Review · Reviewer_XfV5 · 2026-03-18

**Soundness:** 3
**Presentation:** 3
**Significance:** 3
**Originality:** 3
**Overall Recommendation:** 5
**Confidence:** 3

**Summary:**

This paper proposes CURVE, a structured framework for relational modeling that combines uncertainty modeling, scene graph learning, and causality-inspired representation learning. The method aims to address the problem that scene graphs often overfit to spurious correlations, which harms out-of-distribution generalization. Specifically, CURVE integrates variational uncertainty modeling with uncertainty-guided structural regularization to suppress high-variance, environment-specific relations, and uses prototype-conditioned debiasing to disentangle invariant interaction dynamics from environment-dependent variations. The experiments show that the proposed method improves performance over several baselines in out-of-distribution settings and produces sparser and more stable graph structures, indicating potential value for autonomous driving risk prediction tasks.

**Compliance With Llm Reviewing Policy:**

Affirmed.

**Final Justification:**

I thank the authors for their detailed rebuttal. Most of my concerns have been addressed with additional analysis and experiments. While there is still room for further exploration, the current version provides sufficient evidence and clarity. I consider the paper to meet the bar for acceptance and maintain my rating of Accept (5).

**Key Questions For Authors:**

1. Can the authors provide additional evidence to show whether the proposed method generalizes beyond autonomous driving collision risk prediction, for example on other scene graph or visual relationship tasks? A positive answer would strengthen the claim that the method has broader applicability.

2. Can the authors include comparisons with more recent methods related to out-of-distribution generalization or causal representation learning? Such results would help clarify the competitiveness of the proposed method.

3. Can the authors analyze the effect of the number of environmental prototypes on model performance, and discuss whether there is redundancy or semantic interpretability in the learned prototypes? This would improve the credibility and interpretability of the method.

4. Can the authors further evaluate the uncertainty-guided graph pruning strategy against simpler alternatives such as threshold-based pruning? This would help better justify the specific design choice of the proposed pruning mechanism.

**Limitations:**

The limitations could be discussed more explicitly. In particular, the paper should better acknowledge the limited experimental scope, the uncertainty regarding generalization to other scene graph or visual relationship tasks, and the lack of deeper analysis of the environmental prototype mechanism and the pruning strategy.

**Strengths And Weaknesses:**

From an overall perspective, the paper combines uncertainty modeling, scene graph learning, and causality-inspired representation learning to propose a structured approach for relational modeling. The experimental results show that the proposed method achieves performance improvements over several baselines in out-of-distribution scenarios and produces sparser and more stable graph structures, which demonstrates potential value for autonomous driving risk prediction tasks.

However, several issues still need further improvement.

First, the experimental scope is relatively limited. The current evaluation focuses mainly on autonomous driving collision risk prediction and lacks validation on other scene graph or visual relationship tasks. Therefore, the general applicability of the proposed method remains to be further demonstrated.

Second, the baseline comparisons are not sufficiently comprehensive. The current experiments mainly compare with several graph neural network-based models. It would be beneficial to include comparisons with more recent methods related to out-of-distribution generalization or causal representation learning.

Third, the environmental prototype mechanism, which is a key component of the proposed method, lacks deeper analysis. The paper does not systematically investigate the impact of the number of prototypes on model performance, nor does it discuss potential redundancy or semantic interpretability of these prototypes. Additional sensitivity analysis or visualization results would help improve the credibility and interpretability of the method.

Fourth, the ablation studies are still relatively limited. The current analysis mainly removes certain modules for comparison. It would be helpful to further analyze the effectiveness of the uncertainty-guided graph pruning strategy compared with simpler alternatives such as threshold-based pruning.

Overall, the proposed framework is conceptually interesting, but the experimental validation is still somewhat limited. Providing more comprehensive experimental comparisons and a more systematic analysis of key components would further strengthen the paper.

---

> ### Author Rebuttal · Authors · 2026-03-28
>
> We thank Reviewer XfV5 for the constructive feedback and recognition of our promising OOD performance with sparser, more stable graph structures. We provide additional experimental comparisons and a more systematic analysis of the key components.
>
> ### 1. Generalization Beyond Collision Risk Prediction (W1, Q1).
> To test transfer beyond collision prediction, we evaluate CURVE on **trajectory prediction**. We freeze the graph backbones of both RS2G and CURVE (pre-trained on collision prediction), and only fine-tune the temporal LSTM alongside a lightweight 2-layer MLP head to predict future waypoints (step 7) based on historical trajectories (steps 1-4). As shown in Table I, CURVE shows stronger overall transfer performance in both ID and OOD settings, suggesting that the learned graph structure **captures transferable interaction patterns** beyond the original collision prediction task.
>
> Table I. Task Transfer on Trajectory Prediction.
> |Method|ID (FDE/MAE-x/MAE-y)|OOD (FDE/MAE-x/MAE-y)|
> |:---|:---:|:---:|
> |Rule-based|18.94/14.75/9.12|17.05/12.74/8.80|
> |RS2G|10.04/8.46/3.71|14.70/12.82/3.80|
> |CURVE|6.71/5.14/3.30|9.83/7.68/4.45|
> ### 2. Comparisons with Causal / OOD Baselines (W2, Q2).
> To broaden our baselines, we additionally compare CURVE with **IRM**, a representative OOD generalization objective, and **DDLoss**, a recent causality-guided debiasing method. Both methods are implemented under the same dynamic spatio-temporal backbone and evaluation protocol for a controlled comparison. As shown in Table II, CURVE achieves stronger ID and OOD performance.
>
> Table II. Comparisons with Causal/OOD Baselines.
> |Method|ID (Acc/AUC/MCC)|OOD (Acc/AUC/MCC)|
> |:---|:---:|:---:|
> |IRM|90.04/93.94/67.29|60.44/61.60/27.94|
> |DDLoss|89.30/95.51/63.59|63.74/66.76/34.64|
> |CURVE (ours)|92.92/96.19/68.85|68.13/71.15/35.07|
> ### 3. Analysis of Environmental Prototypes (W3, Q3).
> To address your concerns about interpretability and potential redundancy among prototypes, we further analyze the prototype mechanism from two aspects: semantic usage patterns and sensitivity to the prototype number $K$.
> 1. **Semantic Interpretability.** By analyzing the dynamically activated attention weights, we find that the prototypes show **context-dependent specialization without explicit supervision**. For instance, $P_{12}$ is highly active in sparse environments, whereas $P_0$ becomes more active in dense environments and surpasses $P_{12}$. Meanwhile, $P_8$ plays a secondary role, while $P_6$ remains nearly inactive in both cases. These results suggest that the prototypes capture different latent environmental modes rather than behaving as interchangeable slots. The quantitative attention distributions are summarized in Table III, while the qualitative visualizations and the top-5 most strongly activated scenes for $P_{12}$ and $P_0$ are provided in Figures 1 and 2, respectively, in the link https://anonymous.4open.science/r/figures-9AC0/figure.pdf.
>
> Table III. Attention Weight Distribution across Environments.
> |Env|$P_{12}$|$P_0$|$P_8$| $P_6$|
> |:---|:---:|:---:|:---:|:---:|
> |Sparse|0.388|0.305|0.140|0.004|
> |Dense|0.337|0.353|0.162|0.004|
>
> 2. **The Impact of the Number of Prototypes and Redundancy.** We further conduct an ablation on the number of prototypes $K$ in Table IV. CURVE is **relatively stable** across $K \in \\{8,16,32,64\\}$, while $K=32$ gives the best overall ID/OOD trade-off. This indicates that the gains do not rely on a brittle choice of dictionary size. We also observe that some prototypes have low activation in the source domain. Rather than claiming that all prototypes are always equally useful, we interpret this as evidence that the dictionary contains both frequently used prototypes and reserve capacity that may become more relevant under broader domain shifts. Consistent with this, in sim-to-real analysis on DoTA dataset, some weakly activated prototypes in the source domain become more active in real-world scenes (visualized in Figure 3 in the same link).
>
> Table IV. Ablation on Prototype Number($K$).
> |Variant|ID (Acc/MCC)|OOD (Acc/MCC)|Complexity (Avg Deg.)|
> |:---|:---:|:---:|:---:|
> |$K=8$|90.61/55.98|67.03/32.78|5.75|
> |$K=16$|91.28/60.46|68.13/35.62|5.76|
> |$K=32$ (Def.)|92.92/68.85|68.13/35.07|5.78|
> |$K=64$|91.28/59.82|67.03/33.04|5.57|
>
> ### 4. Ablations with Threshold-based Pruning (W4, Q4).
> We clarify that the original manuscript already includes a threshold-based pruning baseline, denoted as **RS2G (thr=0.6)**, which relies on direct weight thresholding without uncertainty guidance. To make the comparison more controlled, we additionally evaluate RS2G (thr=0.7), whose graph density closely matches that of CURVE.
>
> Table V. Comparison of Pruning Strategies.
> |Model|ID(Acc/MCC)|OOD(Acc/MCC)|Complexity(Avg Deg./Edges.)|
> |:---|:---:|:---:|:---:|
> |Thresholding-Pruning (Thr=0.6)|90.32/82.01|57.14/12.91|7.42/78.37|
> |Thresholding-Pruning (Thr=0.7)|86.02/80.06|54.86/12.87|5.64/42.00|
> |CURVE (ours)|92.92/68.85|68.13/35.07|5.78/44.80|

---

> > ### Author Rebuttal · Reviewer_XfV5 · 2026-04-08
> >
> > I thank the authors for their detailed rebuttal. Most of my concerns have been addressed with additional analysis and experiments. While there is still room for further exploration, the current version provides sufficient evidence and clarity. I consider the paper to meet the bar for acceptance and maintain my rating of Accept (5).

---

> > > ### Author Response · Authors · 2026-04-08
> > >
> > > Thank you very much for your careful reconsideration and encouraging feedback. We sincerely appreciate your time and thoughtful comments throughout the review process. We are glad that our rebuttal helped address your concerns. Thank you again for your support!

---

### Official Review · Reviewer_f9V3 · 2026-03-19

**Soundness:** 2
**Presentation:** 2
**Significance:** 2
**Originality:** 3
**Overall Recommendation:** 5
**Confidence:** 2

**Summary:**

This paper addresses OOD generalization in scene graph learning by leveraging uncertainty as a proxy for spuriousness to guide graph sparsification, combined with prototype-driven soft causal intervention to disentangle invariant physical dynamics from environmental confounders. The method achieves strong zero-shot transfer performance with significantly sparser topologies and demonstrates high data efficiency in sim-to-real adaptation.

**Compliance With Llm Reviewing Policy:**

Affirmed.

**Final Justification:**

The paper proposed a novel idea to learn invariant representations, and later in the rebuttal, the authors adequately discussed the functionality and interpretability of different prototypes. Ablation studies are also added. And according to the computation overhead, it's also a practical technique. So I decided to raise the score to 5 (Accept).

**Key Questions For Authors:**

1. Can you discuss more about the dictionaries, like the hyperparameters (may use ablation studies to explore), and what do the 32 prototypes actually represent? Are the performances sensitives to these prototypes (numbers, dimensions... )
2. When will CURVE fail?
3. What's the overall computation overhead of the proposed mechanism? What's the latency impact for real-time deployment?

**Limitations:**

Yes.

**Strengths And Weaknesses:**

Strenths: (1) The idea is great. It repurposes uncertainty from denoising to graph pruning rather than mere prediction smoothing (2) Approaches oracle performance with only 5% real-world annotations in sim-to-real transfer. （3）Clear information-theoretic objective which enhances the interpretability.

Weaknesses: (1) Lack of experiments to show the interpretability. For instance, the dictionary representations are not discussed or explored. (2)Lack of  ablation studies like hyper-parameters or weights. Also no principled way to choose these parameters. (3) The overall scope is limited.

---

> ### Author Rebuttal · Authors · 2026-03-28
>
> We appreciate Reviewer f9V3's positive assessment of our idea, particularly the use of uncertainty for graph pruning and the promising sim-to-real transfer results. To address the remaining questions, we provide the following new analyses and experiments.
> ### 1. What Do the Prototypes Capture (W1, Q1)?
> To address this concern, we analyze prototype attention under different traffic contexts. The results suggest that the prototypes capture **different latent environmental patterns** rather than collapsing to uniform usage. For example, in sparse environments (Figure 2a), $P_{12}$ receives the highest attention. In contrast, in dense environments (Figure 2b), $P_0$ becomes more active and surpasses $P_{12}$. Meanwhile, $P_8$ plays a secondary role in dense scenes, while $P_6$ remains nearly inactive. The quantitative attention distributions are summarized in Table I. Qualitative activation distribution visualizations in Figure 1 and the top-5 most strongly activated scenes for $P_0$ and $P_{12}$ in Figure 2 are in https://anonymous.4open.science/r/figures-9AC0/figure.pdf.
>
> Table I. Attention Weight Distribution across Environments.
> |Env|$P_{12}$|$P_0$|$P_8$|$P_6$|
> |:---|:---:|:---:|:---:|:---:|
> |Sparse|0.388|0.305|0.140|0.004|
> |Dense|0.337|0.353|0.162|0.004|
> ### 2. Ablation Studies on Hyperparameters (W2, Q1).
> We evaluate sensitivity to prototype size ($K$), dimension ($d$) and $\lambda_{unc}$, with all other settings fixed.
>
> Table II. Ablation on Prototype Size ($K$).
> |Variant|ID (Acc/MCC)|OOD (Acc/MCC)|Complexity (Avg Deg.)|
> |:---|:---:|:---:|:---:|
> |$K=8$|90.61/55.98|67.03/32.78|5.75|
> |$K=16$|91.28/60.46|68.13/35.62|5.76|
> |$K=32$ (Def.)|92.92/68.85|68.13/35.07|5.78|
> |$K=64$|91.28/59.82|67.03/33.04|5.57|
>
> Table III. Ablation on Prototype Dimension ($d$).
> |Variant|ID (Acc/MCC)|OOD (Acc/MCC)|Complexity (Avg Deg.)|
> |:---|:---:|:---:|:---:|
> |$d=32$|91.47/66.78|67.03/32.83|5.85|
> |$d=64$ (Def.)|92.92/68.85|68.13/35.07|5.78|
> |$d=128$|91.18/60.26|67.03/32.77|5.58|
>
> Table IV. Ablation on Uncertainty Loss Weight ($\lambda_{unc}$).
> |Variant|ID (Acc/MCC)|OOD (Acc/MCC)|Complexity (Avg Deg.)|
> |:---|:---:|:---:|:---:|
> |$\lambda_{unc}=0.1$|88.98/64.83|60.43/22.93|5.80|
> |$\lambda_{unc}=0.5$|91.09/59.05|65.93/30.73|5.85|
> |$\lambda_{unc}=1.0$ (Def.)|92.92/68.85|68.13/35.07|5.78|
> |$\lambda_{unc}=2.0$|90.61/59.44|68.13/36.02|5.77|
>
> Tables II–IV show two patterns. First, CURVE is **relatively stable to $K$ and $d$**: varying the dictionary size or dimension causes only modest OOD changes, suggesting the gains do not rely on brittle choices. Second, $\lambda_{unc}$ should not be too small: 0.1 and 0.5 hurt OOD performance, while 1.0 and 2.0 are similar. We therefore use $K=32$ for the best overall ID/OOD trade-off, $d=64$ to match the RGCN hidden dimension, and $\lambda_{unc}=1.0$ as default since larger values bring no consistent gain.
> ### 3. When will CURVE Fail (Q2)?
> CURVE mainly fails in two cases:
> 1. **Upstream Perception Blind Spots**. Like most graph-based methods, we are fundamentally bound by our upstream detector (Appendix B.1). If a critical object is missed, e.g., due to severe occlusion, no corresponding node exists in the graph.
> 2. **Over-pruning Inherently Unpredictable Risks**. Our sparsification heuristic assumes that high-variance relations are more likely to be environmental noise. However, this assumption can fail when a truly causal relation is itself inherently unpredictable (e.g., an erratic pedestrian). In such cases, the model may mistakenly prune a vital connection, leading to missed risks.
> ### 4. Computation Overhead and Latency Impact (Q3).
> For fair comparison, both RS2G and CURVE are evaluated on pre-extracted scene-graph sequences, excluding upstream detection. Latency is measured from sequence input to final risk prediction on an RTX 3070 GPU.
>
> Table V. Computational Efficiency Analysis.
> |Model|Latency(ms)|FPS|Params(M)|FLOPs(G)|
> |:---|:---:|:---:|:---:|:---:|
> |RS2G|6.25|159.93|0.0191|0.0005|
> |CURVE|13.12|76.2|0.0231|0.0012|
>
> While CURVE adds relative overhead, the absolute increment is only +0.004M Params and +6.87 ms. Achieving 76.2 FPS, CURVE satisfies real-time autonomous driving requirements (10-30 FPS).
> ### 5. Regarding the Overall Scope (W3).
> To further test transfer beyond risk assessment, we evaluate CURVE on an additional downstream **trajectory prediction** task. We freeze the graph structural backbones of RS2G and CURVE pre-trained on collision prediction, and only fine-tune the temporal LSTM with a lightweight 2-layer MLP head to predict future waypoints (step 7) from historical trajectories (steps 1–4). CURVE shows better overall transfer on both ID and OOD settings, providing additional evidence for its **task-agnostic design**.
>
> Table VI. Task Transfer on Trajectory Prediction.
> |Method|ID (FDE/MAE-x/MAE-y)|OOD (FDE/MAE-x/MAE-y)|
> |:---|:---:|:---:|
> |Rule-based|18.94/14.75/9.12|17.05/12.74/8.80|
> |RS2G|10.04/8.46/3.71|14.70/12.82/3.80|
> |CURVE|6.71/5.14/3.30|9.83/7.68/4.45|

---

> > ### Author Rebuttal · Reviewer_f9V3 · 2026-04-04
> >
> > Thank the reviewers for adequate supplemental experiments. My questions are fully resolved.

---

> > > ### Author Response · Authors · 2026-04-04
> > >
> > > Thank you very much for your follow-up and for acknowledging the supplemental experiments! We are glad to know that your questions have been fully resolved, and we appreciate your consideration in the final evaluation.

---

### Decision · Program_Chairs · 2026-04-30

**Decision:**

Accept (regular)

**Comment:**

This paper introduces a method that combines soft causal intervention with uncertainty-guided structural regularization to learn sparse scene graphs by suppressing environment-specific spurious relations induced by environmental confounders. Experimental results show that the proposed approach learns domain-invariant scene graphs and improves performance over several baselines under distribution shifts.  Reviewers highlight several strengths, including a clear information-theoretic objective that unifies causal representation learning, uncertainty modeling, and scene graph learning, as well as strong data efficiency on real-world datasets. The main concerns center on insufficient experimental validation, including limited baseline comparisons, a lack of comprehensive ablation studies, and evaluation on only a single downstream task.  The authors’ rebuttal adequately addresses these concerns; in particular, the additional experiments significantly strengthen the paper. The final ratings are two accepts and one weak accept.